# Long-term Assessment of Primary and Secondary Organic Aerosols in Shanghai Megacity throughout China's Clean Air Actions since 2010

Haifeng Yu[1], Yunhua Chang[1, *], Lin Cheng[1], Yusen Duan[2], and Jianlin Hu[3, *]

[1]Collaborative Innovation Center on Forecast and Evaluation of Meteorological Disasters, NUIST Center on Atmospheric Environment, Nanjing University of Information Science & Technology (NUIST), Nanjing 210044, China
[2]Shanghai Environmental Monitoring Center, Shanghai 200235, China
[3]Jiangsu Key Laboratory of Atmospheric Environment Monitoring and Pollution Control, Collaborative Innovation Center of Atmospheric Environment and Equipment Technology, Nanjing University of Information Science & Technology, Nanjing 210044, China

*Correspondence to*: Yunhua Chang (changy13@nuist.edu.cn; ORCID: 0000-0002-1622-5330) and Jianlin Hu (jianlinhu@nuist.edu.cn; ORCID: 0000-0002-5694-4794)

**Abstract.** A growing body of research has demonstrated the effectiveness of China's Air Pollution Prevention and Control Action Plan in controlling $PM_{2.5}$ pollution. However, there is a lack of long-term studies investigating the impact of these abatement policies on carbonaceous aerosols in $PM_{2.5}$, particularly secondary organic carbon (SOC). Shanghai, as China's largest megacity and prominent industrial hub, serves as a crucial gateway to the nation's rapid development with a population exceeding twenty million. In this study, we conducted hourly online measurements of organic carbon (OC) and elemental carbon (EC) in $PM_{2.5}$ in Shanghai from July 2010 to July 2017. The results revealed that the annual concentrations (mean $\pm$ 1 $\sigma$) of OC and EC reached their peaks in 2013 ($9.5 \pm 6.4$ and $2.7 \pm 2.6$ $\mu g$ $m^{-3}$ to $3.0 \pm 2.3$ $\mu g$ $m^{-3}$ and $2.7 \pm 2.1$ $\mu g$ $m^{-3}$). Subsequently, a consistent year-by-year decrease in both OC and EC concentrations was observed, mirroring the trend observed for $PM_{2.5}$. Primary organic carbon (POC), the primary component of OC, accounted for an average of 65.6%, displaying similar trends to OC. This finding indicates the effectiveness of primary emission control measures. However, the concentration of secondary organic carbon (SOC) did not decrease from 2013 to 2017, remaining relatively stable within the range of $2.7 \pm 2.6$ $\mu g$ $m^{-3}$ to $3.0 \pm 2.3$ $\mu g$ $m^{-3}$. When considering data from previous studies in Shanghai, concentrations of SOC did not exhibit a noticeable decline until 2018, coinciding with the implementation of measures targeting volatile organic compounds (VOCs) emissions. Seasonally, with the exception of 2011, OC and EC concentrations were highest during winter, likely influenced by unfavourable meteorological conditions and long-range transport. SOC displayed no distinct seasonal fluctuations, as its formation is influenced by both photochemical reactions and meteorological conditions. POC and SOC exhibited different diurnal patterns, but neither showed a significant weekend effect, suggesting limited reduction in anthropogenic activities during weekends. Furthermore, SOC concentrations exhibited simultaneous increases in summer, particularly when $O_3$ concentrations exceeded 50 $\mu g$ $m^{-3}$, indicating that stronger oxidation reactions contribute to higher SOC concentrations. Our findings also revealed concentration gradients of SOC dependent on wind direction (WD) and wind speed

(WS), with higher concentrations typically observed for winds originating from the southwest and northwest. Potential sources from distant regions were analyzed using the potential source contribution function (PSCF), indicating that the geographical potential source area is concentrated near the middle and lower Yangtze River.

## 1 Introduction

Carbonaceous aerosols significantly contribute to $PM_{2.5}$ (particulate matter with an aerodynamic diameter $\leq$ 2.5 μm), accounting for approximately 10-70% of its mass concentration (Kanakidou et al., 2005; Hu et al., 2012; Tiwari et al., 2016; Ke et al., 2023). Their presence in the atmosphere has profound implications for air quality, climate, and human health, rendering them a subject of global concern (Chung, 2002; Pöschl, 2005; Highwood and Kinnersley, 2006; Cao et al., 2012; Liu et al., 2023). Specifically, elevated levels of carbonaceous aerosols can impair air visibility, a key indicator of urban air quality (Park et al., 2003; Ram, 2022). They also have the potential to modify the climate through directly absorbing solar radiation and indirectly influencing cloud formation and precipitation (Chung et al., 2012). Furthermore, carbonaceous aerosols often contain polycyclic aromatic hydrocarbons (PAHs), which are recognized carcinogens, mutagens, and teratogens, posing significant health risks (Boström et al., 2002; Chen and Liao, 2006; Mallah et al., 2022). Typically, carbonaceous aerosols are classified into three categories: organic carbon (OC), elemental carbon (EC, akin to black carbon (BC)), and carbonate carbon (CC) (Mader et al., 2004; Zhou et al., 2012). OC includes both primary organic carbon (POC), emitted directly from sources such as vehicle exhaust, biomass burning, and cooking, as well as secondary organic carbon (SOC), formed through chemical reactions such as photochemical oxidation initiated by OH radicals, aqueous phase oxidation, and gas/particle partitioning (Turpin and Huntzicker, 1991; Fu et al., 2012b). Similar to POC, EC is predominantly generated through incomplete combustion of biomass and fossil fuels (Husain et al., 2007; Yan et al., 2019). Carbonate carbon, mainly derived from natural mineral dust and dust generated by building and demolition activities, is frequently disregarded in studies due to its minor contribution (Mader et al., 2002).

China is a major emitter of POC and EC, largely driven by substantial energy consumption and a growing vehicle population, which have accompanied rapid economic development and urbanization over the past few decades (http://www.stats.gov.cn/sj/ndsj/2021/indexch.htm). Historically, studies on carbonaceous aerosols in China were limited, with early investigations by Dod et al. (1986) and Weihan et al. (1987) in Beijing identifying coal combustion as the primary source of these aerosols. As analytical techniques have progressed from offline filter sampling to online measurements, the understanding of carbonaceous aerosols has become more comprehensive, resulting in a proliferation of studies across various Chinese cities. These include Beijing (Zhang et al., 2007; Ji et al., 2018; Hou et al., 2021), Shanghai (Fu et al., 2012a; Wang et al., 2016b; Wei et al., 2019), Nanjing (Chen et al., 2017; Liu et al., 2019; Dai et al., 2022), Xi'an (Li et al., 2008; Han et al., 2016; Ni et al., 2018), Chengdu (Chen et al., 2014; Shi et al., 2016; Xiang et al., 2021), and Guangzhou (Duan et al., 2007; Liu et al., 2014; Cheng et al., 2021). Notably, Cao et al. (2007) conducted a national-scale measurement of atmospheric OC

and EC in 14 Chinese cities during the winter and summer of 2003. Their findings demonstrated that OC and EC concentrations exhibit summer minima and winter maxima across all cities.

In response to severe air pollution, China implemented an air pollution prevention and control action plan in 2013. This initiative has been inducing significant changes in atmospheric composition, including volatile organic compounds (VOCs), nitrogen oxides ($NO_x$), and ozone ($O_3$), which may substantially impact SOC formation. Shanghai, as the largest economic center with the most extensive urban population in China, has pioneered comprehensive air pollution control measures. Long-term observations of OC and EC are crucial for evaluating the effectiveness of these policies in reducing carbonaceous aerosol pollution. While Wu et al. (2019) and Xu et al. (2018) conducted continuous observations of SOC in Guangzhou and Shanghai, respectively, their studies were limited to one year. Ji et al. (2019) and Chang et al. (2017) provided long-term observations in Beijing and Shanghai, but their studies did not cover the period both before and after the 2013 action plan implementation nor provide SOC estimations. Wang et al. (2022) analyzed SOC in Shanghai from 2016 to 2020 but focused on non-urban areas and did not compare characteristics before and after 2013. Therefore, there is a need to understand the long-term impacts of air pollution control measures on SOC in urban settings.

Addressing these research gaps, this study investigates the long-term evolution of carbonaceous aerosols in Shanghai, emphasizing variations before and after the 2013 emission reduction policy. Using a Sunset Laboratory online OC/EC analyzer, we obtained high-resolution, hourly data on OC and EC from an urban-representative site. The study also examines the influence of meteorological conditions and other pollutants on SOC formation across different seasons and analyzes potential SOC sources. The primary objective is to enhance understanding of the variability and sources of SOC. The long-term data generated through this research provide essential insights for assessing the effectiveness of current air pollution control policies and guiding future strategies to mitigate pollution.

## 2 Materials and Methods

### 2.1 Observation site

The sampling site for this study is located on the open-top terrace of the Pudong Environmental Monitoring Center office building, 18 m above ground level, in downtown Shanghai (121.5446°E, 31.2331°N; Fig. S1), approximately 5 km from the city center. This site represents a typical urban setting with a mix of office complexes, commercial establishments, residential neighborhoods, and active traffic routes. The absence of tall structures and industrial facilities within a radius of at least 1 km significantly enhances the accuracy of airborne particulate matter measurements. Furthermore, measurements are conducted on the rooftop platform, ensuring that there are no obstructions from overhead structures. Previous research has confirmed the suitability of this location for examining air quality dynamics in the Yangtze River Delta region (Chang et al., 2016).

## 2.2 Instrumentation

From July 10, 2010, to July 31, 2017, we conducted online observations of ambient concentrations of OC and EC using a Sunset Laboratory semi-continuous OC/EC analyzer (RT-4 model, Sunset Laboratory Inc., USA). In the measurement process, ambient air was initially drawn into the instrument at a flow rate of 8 L min$^{-1}$. After traversing a multichannel, parallel plate denuder equipped with a carbon-impregnated filter to eliminate semi-volatile organics, the aerosols were collected on a quartz filter. Subsequently, the collected aerosols (OC and EC) underwent analysis for thermo-optical transmittance (TOT) in a sample oven following the NIOSH method 5040. To maintain accuracy, we replaced the quartz filters every 3-5 days and conducted routine instrument blank tests. The relative standard deviations of measurement uncertainties for OC and EC remained below 5%, with detection limits of 0.2 µg m$^{-3}$ and 0.04 µg m$^{-3}$ for OC and EC, respectively. Further details regarding related instrumentation can be found elsewhere (Chang et al., 2017).

Additionally, the mass concentration of PM$_{2.5}$ was determined using a Thermo Fisher Scientific TEOM 1405-D. Hourly concentrations of carbon monoxide (CO), nitrogen dioxide (NO$_2$), sulfur dioxide (SO$_2$), and ozone (O$_3$) at the site were provided by the Pudong Environmental Monitoring Center, with all instruments located on the rooftop. Meteorological parameters, including ambient temperature (T), relative humidity (RH), wind speed (WS), and wind direction (WD), were sourced from the local meteorological office located approximately 1 km to the southeast (refer to supplementary information for the site map). All online measurements mentioned above were recorded at a 1-hour resolution.

## 2.3 Method for estimating secondary organic carbon

Various calculations have been proposed to estimate the concentrations of SOC and POC. EC, derived exclusively from primary emissions and minimally chemically transformed in the atmosphere, serves as an ideal tracer for POC. The formula for estimating SOC concentration is (Turpin et al., 1995; Wang et al. 2022):

$$OC_{tt} = POC + SOC \qquad (1)$$
$$POC = EC \times (OC / EC)_{pri} \qquad (2)$$
$$SOC = OC_{tt} - POC \qquad (3)$$

where OC$_{tt}$ represents the measured organic carbon concentration, and (OC / EC)$_{pri}$ denotes the OC/EC ratio specific to aerosols from predominant emission sources. Given that the OC/EC ratio can vary significantly due to diverse emission sources and meteorological conditions, establishing a reliable (OC / EC)$_{pri}$ ratio is crucial for the EC tracer method. Therefore, the minimum correlation coefficient (MRS) method was employed in this study to estimate SOC. In a prior investigation, Wu et al. (2018) compared different methodologies for determining the estimated (OC / EC)$_{pri}$ through a numerical analysis. Their findings indicated that the MRS method exhibits greater robustness and minimal computational errors compared to the minimum OC/EC and percent OC/EC methods in SOC estimation. In the MRS approach, EC solely originates from primary sources, and the correlation coefficient $R^2$ is computed using measured EC values and SOC estimates derived from a range of

hypothetical $(OC / EC)_{pri}$ ratios. The OC/EC ratio corresponding to the minimum $R^2$ is determined by plotting $R^2$ against the corresponding hypothetical $(OC / EC)_{pri}$ ratio. This identified ratio aligns with the actual $(OC / EC)_{pri}$ ratio. Instances where the measured OC/EC ratio falls below the estimated $(OC / EC)_{pri}$ ratio yield a negative SOC value (Wu and Yu, 2016). In handling this uncertainty, it is assumed that there is no secondary formation in the analyzed OC samples. Given that the relative contributions of major emission sources vary across different months, the $(OC / EC)_{pri}$ ratio was calculated for each month of the observation period using the MRS method.

## 2.4 Geographical origin analysis

In this study, we employed bivariate pole plots (BPPs) and potential source contribution functions (PSCFs) to delineate both intra-city transport patterns and long-range geographic source regions of SOC in Shanghai, respectively.

BPPs enable the visualization of the synergistic variation of WS and WD in polar coordinates alongside pollutant concentrations. Previous research has demonstrated BPPs' efficacy in discerning various emission source areas. Initially, WD, WS, and pollutant concentration data are binned according to wind speed direction, and the mean pollutant concentration is calculated for each bin. Wind vectors are calculated using the following equations (Monahan et al., 2018):

$$u = u'.\sin(2\pi / \theta) \qquad (4)$$
$$v = u'.\cos(2\pi / \theta) \qquad (5)$$

where $u'$ represents the average wind speed per hour, $\theta$ denotes the average wind direction in degrees, and 90 degrees signifies the east wind. Subsequently, the function constructs a 3D surface incorporating WS, WD, and pollutant concentration (C). To enhance air pollution prediction, we introduced a generalized additive model (GAM) framework. Given the nonlinear relationships between variables, GAMs account for interactions among these variables. Further details regarding this model can be found in the R "openair" toolkit.

The PSCF method was employed to explore the regional source areas of air pollution. We calculated the 72-hour backward trajectory of SOC for each of the four seasons at a sampling point located 500 m above ground level. These trajectories were computed utilizing the NOAA Hybrid Single Particle Lagrangian Integrated Trajectory (HYSPLIT5.2) model and meteorological data sourced from the Global Data Assimilation System (ftp://ftp.arl.noaa.gov/pub/archives/gdas1). The model run interval was set to 1 hour. The PSCF is expressed as follows (Bressi et al., 2014):

$$PSCF(i, j) = w_{ij} \times (m_{ij} / n_{ij}) \qquad (6)$$

here $(i, j)$ the number of trajectory endpoints falling into a specific grid cell, indicating the collection of material emissions from that cell; $n_{ij}$ denotes the total number of trajectory endpoints associated with aerosol species concentrations surpassing a defined threshold. A higher PSCF value indicates a greater potential source contribution to the receptor site. In our study, the

PSCF domain was established within a grid cell encompassing all backward trajectories (with an accuracy of 0.1° × 0.1°), and the 75th percentile value of SOC for each season served as the threshold $m_{ij}$. This method is executed in the TrajStat toolbox of MeteInfo (Carslaw and Ropkins, 2012).

## 3 Result and Discussion

### 3.1 Overview of carbonaceous aerosol data

Table 1 presents a comprehensive overview of the concentrations of carbonaceous aerosol species, including EC, OC, POC, and SOC, over the study period. Data availability for OC and EC was 63.8%, with some variation across different years and seasons. In line with the conventional Chinese calendar, each year spans from March of one year to February of the following year. For instance, the year 2011 encompasses March 2011 to February 2012. Seasons were classified as follows: spring (March to May), summer (June to August), autumn (September to November), and winter (December to February).

Figure 1 displays the long-term trends in carbonaceous aerosols and $PM_{2.5}$ in Shanghai. Through extending the data reported by Wang et al. (2022) from 2016 to 2020, we covered a decade-long period from 2010 to 2020. The annual average concentration of $PM_{2.5}$ peaked in 2013 at $59.5 \pm 37.8 \, \mu g \, m^{-3}$ and reached its lowest in 2020 at $30.2 \pm 14.0 \, \mu g \, m^{-3}$. Carbonaceous aerosols (calculated as the sum of OC multiplied by a factor of 1.4 (Turpin et al., 2001) and EC) contributed an average of $22.3\% \pm 13.0\%$ to the total $PM_{2.5}$ mass, underscoring their significant role. The total carbon (TC), which is the sum of OC and

EC, also showed a peak in 2013 ($12.2 \pm 8.3 \, \mu g \, m^{-3}$) and a low in 2020 ($6.0 \pm 3.5 \, \mu g \, m^{-3}$). Primary carbonaceous aerosol concentrations (POC and EC) were highest in 2013 ($6.8 \pm 5.3 \, \mu g \, m^{-3}$ and $2.7 \pm 2.1 \, \mu g \, m^{-3}$, respectively) and lowest in 2020 ($3.6 \pm 2.3 \, \mu g \, m^{-3}$ and $1.0 \pm 0.6 \, \mu g \, m^3$, respectively). SOC concentrations varied slightly from 2013 to 2017 ($2.7 \pm 2.6 \, \mu g \, m^{-3}$ to $3.0 \pm 2.3 \, \mu g \, m^{-3}$) and started to decrease post-2018. To minimize errors in SOC estimation, the MRS method was used to calculate monthly $(OC / EC)_{pri}$ ratios (see Fig. S2-S9), ranging from 1.0 to 3.3, aligning with previous studies using the same

method (Wang et al., 0.9-4.7; Xu et al., 1.0-2.2).

Figure 1a shows a significant reduction in $PM_{2.5}$ levels in Shanghai, with a 50.7% decrease from 2013 to 2020, and a 41.7% decrease from 2013 to 2019. This trend mirrors reductions seen in other major regions of China, such as Beijing-Tianjin-Hebei, the Yangtze River Delta, and the Pearl River Delta, where $PM_{2.5}$ concentrations fell by 15-25% between 2013 and 2017 (Gao et al., 2018; Dai et al., 2021; Yan et al., 2020). This decline in annual average concentrations is likely due to the implementation

of the Air Pollution Prevention and Control Action Plan. It is important to note that the year 2020 was particularly influenced by the COVID-19 pandemic, including it as a comparison baseline may overestimate the reduction in $PM_{2.5}$ concentrations. Therefore, it is more appropriate in our study to focus on the trend analysis before the pandemic began. Additionally, it is important to understand that PM2.5 concentrations are influenced not only by emission control measures but also significantly by meteorological conditions. Shanghai is located in a subtropical monsoon climate zone, where interannual variations in

meteorological conditions (such as wind direction, temperature inversion, and boundary layer height) play a crucial role in the dispersion and transformation processes of pollutants. These meteorological factors are reflected in the observed changes in carbonaceous aerosol concentrations. Previous studies have shown that $PM_{2.5}$ concentrations are influenced by both anthropogenic emissions and meteorological conditions, with the latter having a limited impact on emission reductions (Chen et al., 2019; Chen et al., 2020). In Fig. 1a, the observed decrease in primary carbonaceous aerosols (POC + EC) in $PM_{2.5}$ can

be attributed to various emission control measures, such as closing small industrial boilers, cleaning large industrial boilers, phasing out all "yellow label" cars, and replacing coal with clean energy. Furthermore, the potential source contribution function analysis emphasizes the impact of regional transport under specific wind directions, which may lead to variations in carbonaceous aerosol concentrations across different years. The boundary layer height and temperature inversion phenomena during winter also enhance the retention of pollutants, further affecting the concentrations of carbonaceous aerosols.

Considering the significant impact of VOCs on the formation of secondary organic aerosols, it is possible that the lack of decrease in SOC concentrations from 2013 to 2017 (Fig. 1a) may have been influenced by insufficient control of VOC emissions (Tsigaridis and Kanakidou, 2007; Simayi et al., 2022). However, the situation regarding VOCs in China has improved in recent years, with increased attention to their regulation. Recently, there has been a push to control VOC emissions from sintering flue gases. In 2020, the Ministry of Ecology and Environment of the People's Republic of China revised the air

pollution evaluation index, shifting the focus from "$SO_2$ and $NO_x$" to "VOCs and $NO_x$". Stringent control of VOC emissions in recent years has led to the first signs of a decrease in SOC starting in 2018 (starting in 2018; https://www.mee.gov.cn/ywgz/fgbz/gz/201807/t20180705_446146.shtml). This demonstrates the effectiveness of emission control measures in reducing carbonaceous aerosols.

Figure 1b illustrates the mass fractions of carbonaceous aerosols across various $PM_{2.5}$ concentration levels throughout the

study period to investigate their contribution to elevated $PM_{2.5}$ pollution. Air pollution levels are classified as Excellent ($0 < PM_{2.5} \leq 35$ μg m$^{-3}$), Good ($35 < PM_{2.5} \leq 75$ μg m$^{-3}$), Light pollution ($75 < PM_{2.5} \leq 100$ μg m$^{-3}$), Moderate pollution ($100 < PM_{2.5} \leq 200$ μg m$^{-3}$), and Heavy pollution ($PM_{2.5} > 200$ μg m$^{-3}$), in accordance with Chinese air quality standards for Class I and Class II. The data indicate that concentrations of POC, SOC, and EC increase with the degradation of air quality. Specifically, the concentrations (in μg m$^{-3}$) of POC, SOC, and EC for each pollution level were: Excellent (2.7, 1.9, 1.3), Good

(4.7, 2.5, 2.3), Light Pollution (7.1, 3.0, 3.4), Moderate Pollution (10.8, 3.6, 5.1), and Heavy Pollution (15.3, 3.4, 6.7). However, the percentages of POC, SOC, and EC in relation to $PM_{2.5}$ decrease as air quality worsens. As air quality declines, the proportion of primary carbonaceous aerosols (POC + EC) in $PM_{2.5}$ decreases from 19.3% to 7.7%, and the proportion of SOC decreases from 9.2% to 1.9%. These findings suggest that as $PM_{2.5}$ levels increase, other constituents besides TC become more significant contributors to the pollution. This is consistent with previous studies demonstrating that secondary inorganic ions

critically contribute to the rise in $PM_{2.5}$ concentrations (Ji et al., 2014; Qiao et al., 2019). To delve into the role of carbonaceous aerosols in PM2.5 pollution and their long-term evolution, this study systematically analyzed the impact of the Air Pollution Prevention and Control Action Plan (implemented in 2013) on PM2.5 and its components, based on long-term observational

data from 2010 to 2020. By integrating hourly online monitoring data of POC, SOC, and EC with meteorological data, a high-resolution dataset was constructed to meticulously depict the seasonal and diurnal variations of primary and secondary carbonaceous aerosols. Combined with Shanghai's pollution control practices, this study further quantified the combined effects of regional transport and local emissions on SOC and POC levels, providing a scientific basis for the formulation of precise prevention and control strategies in the future, as well as offering references for pollution management in other similar regions.

## 3.2 Monthly and seasonal variations

The seasonal and monthly characteristics of carbonaceous aerosols are detailed as follows. Figure 2 presents the monthly mean concentrations of OC (Fig. S11), EC (Fig. S12), POC (Fig. S13), and SOC over a seven-year period. The data indicate that OC and EC levels exhibit similar seasonal variations (Fig. S10), with higher mean concentrations during the cold season (November to February) and lower concentrations during the warm season (March to October). Elevated levels of OC and EC during the colder months are attributed to increased fuel consumption for domestic heating and unfavourable meteorological conditions such as lower mixing layer height, temperature inversions, and calm winds (Cao et al., 2003; Zhao et al., 2013). Conversely, lower levels of OC and EC during the warm season can be attributed to the absence of heating-related emissions, frequent precipitation resulting in strong wet scavenging, and more unstable atmospheric conditions promoting pollutant dispersion. POC shows similar variations to OC, accounting for 65.4% on average of OC. The highest average POC concentration was $7.1 \pm 5.3$ μg m$^{-3}$ in December, while the lowest was $2.7 \pm 1.8$ μg m$^{-3}$ in September. The highest average SOC concentration was observed in January ($3.0 \pm 2.8$ μg m$^{-3}$), and the lowest in October ($2.0 \pm 1.9$ μg m$^{-3}$).

As shown in Fig. 3, the concentrations of carbonaceous aerosols exhibit distinct seasonal variations. The mean concentrations of EC ranged from $1.2 \pm 0.8$ μg m$^{-3}$ in the autumn of 2015 to $3.6 \pm 2.7$ μg m$^{-3}$ in the winter of 2013. Similarly, the mean concentrations of OC varied from $4.4 \pm 2.7$ μg m$^{-3}$ in the spring of 2012 to $12.0 \pm 7.9$ μg m$^{-3}$ in the winter of 2013. Both EC and OC, along with their subtypes POC and SOC, generally showed higher concentrations in winter and lower concentrations in other seasons, except for the year 2011. The elevated winter concentrations of carbonaceous aerosols are influenced by both local emissions and regional transport. During winter, unfavorable meteorological conditions, such as temperature inversions and low mixing layer heights, enhance the trapping of local emissions, including those from domestic heating and biomass burning. At the same time, regional transport from neighboring areas, particularly the Yangtze River Delta, also contributes to the increased levels of carbonaceous aerosols. This is especially significant in winter when long-range transport of pollutants is more common due to prevailing wind patterns. While primary organic carbon (POC) emissions in Shanghai are predominantly from local sources, the regional transport of secondary organic aerosols (SOC) and other pollutants also plays a role in the observed winter concentrations. Therefore, the elevated winter concentrations of carbonaceous aerosols are the result of a combination of both local and regional factors.

A consistent pattern emerged in the seasonal variations of POC concentrations, revealing lower levels in spring, summer, and autumn compared to winter. This trend suggests that POC emissions in Shanghai are predominantly from local sources. Specifically, POC concentrations in winter were 5.8 µg m$^{-3}$, representing 1.25, 1.31, and 1.30 times higher levels than those in spring, summer, and autumn, respectively. The average concentrations of SOC ranged from $1.5 \pm 1.4$ µg m$^{-3}$ (in autumn 2012) to $3.9 \pm 2.2$ µg m$^{-3}$ (in spring 2016) during the study period. Unlike POC, no clear seasonal trend was observed in the changes of SOC over the seven-year period, reflecting the complexity of its formation processes. SOC concentrations are influenced by both primary emissions aging and secondary formation from precursor gases. The precursors for SOA include VOCs, which can be emitted from both anthropogenic and biogenic sources. In the context of the Yangtze River Delta, anthropogenic VOCs are mainly emitted from sources such as vehicle exhaust, industrial activities, and solvent use (An et al., 2021), while biogenic VOCs, including isoprene and terpenes, are emitted by vegetation. These VOCs undergo photochemical oxidation in the atmosphere, leading to the formation of SOA. In addition to VOCs, NO$_x$ and O$_3$ also play critical roles in the formation of SOC. The presence of NO$_x$, primarily from vehicular emissions and industrial activities, contributes to the oxidation of VOCs, facilitating the formation of SOC. The photochemical reactions involving VOCs and O$_3$ are particularly important in the summer months, when intense solar radiation enhances these processes (Wang et al., 2016a). In the region of interest, the primary sources influencing SOC formation include local vehicular emissions, industrial processes, biomass burning, and regional transport of precursor gases. During the winter months, when biomass burning for heating is prevalent, emissions from this source significantly contribute to the formation of SOC, in addition to the more typical urban emissions. The combination of these factors results in the formation and accumulation of SOC, especially during periods of favorable conditions for photochemical reactions and when meteorological conditions trap pollutants near the surface. Consequently, SOC concentrations can exhibit significant variations across seasons.

## 3.3 Diurnal variation and weekly pattern for POC and SOC

Figure 4a depicts the diurnal variations of SOC and POC. SOC exhibits a daily nadir during the morning rush hours (around 08:00 local time), representing a reduction of approximately 35.7% from the midday zenith (around 13:00 local time). Conversely, POC reaches its zenith concentration during the morning commute hours, indicating a significant contribution from local primary vehicular emissions. This observation aligns with prior roadside investigations in other megacities, such as Hong Kong (Huang et al., 2014b), where vehicular emissions were found to have a substantial influence. Notably, while vehicular emissions are the primary contributor to the morning peak in POC, household heating also plays an important role in sustaining POC levels over a 24-hour period, particularly during winter. Following the morning nadir, SOC concentrations surge sharply, reaching a peak around noontime due to increased solar radiation and temperature. Subsequently, both SOC and POC display a descent owing to the progression of the boundary layer and enhanced vertical mixing until the evening rush hour. The magnitude of the evening peaks for SOC and POC is slightly lower than that of the morning zenith, partly due to the higher boundary layer height and wind speed during the evening period. Moreover, the prolonged duration of the evening rush hour leads to diminished vehicle emission rates. Considering the constrained photochemical activity during the evening

rush hour, the evening peak in SOC can be attributed to both intensified emissions and unfavourable meteorological conditions, such as a shallow planetary boundary layer height. A precedent study delineated the diurnal variation of PBLH across different latitudes and surface features based on 45 years (1973-2017) of radiosonde observations (Gu et al., 2020). The findings revealed a markedly lower PBLH during the nocturnal period (6:00 to 8:00 pm) compared to midday (12:00 to 2:00 pm). The confluence of the reduced PBLH and heightened emissions from human activities, such as cooking and commuting, during the evening rush hour, constituted pivotal factors contributing to the observed evening peak in SOC concentrations.

Notably, POC and SOC exhibit distinct weekly patterns. Specifically, POC concentrations were elevated on weekdays compared to weekends during spring and summer, whereas they were higher on weekends than weekdays in autumn and winter. Conversely, SOC concentrations showed a slight increase on weekends relative to weekdays during winter, while the reverse trend was observed in other seasons, as depicted in Fig. 4b. This discrepancy suggests a lack of significant reduction in anthropogenic activity on weekends compared to weekdays, possibly attributable to the absence of license plate-based driving restrictions during weekends (Wang et al., 2022). Further investigation into the sources of SOC and the influence of meteorological parameters on its formation will be explored below.

## 3.4 Insight into SOC formation pathways

EC and a substantial portion of OC directly originate from source emissions, such as fossil fuel combustion and biomass burning, while SOC forms through atmospheric reactions involving $O_3$ and other pollutants. Consequently, long-term and simultaneous measurements of SOC, $SO_2$, $NO_2$, and $O_3$ are essential for understanding emission characteristics and formation processes. Figure 5 illustrates the relationship between SOC and the concentrations of $SO_2$, $NO_2$, and $O_3$. Throughout the study period, Figure 5a shows that $SO_2$ levels consistently declined, indicating effective control measures. The primary source of SO2 emissions in Shanghai is coal combustion, mainly in the power generation sector. Significant control measures, such as the implementation of ultra-low emission technologies in coal-fired power plants since around 2007, have led to a dramatic decrease in SO2 emissions nationwide (Tang et al., 2019). The relatively stable $NO_x$ concentrations could be due to the continuous rise in vehicle numbers, counteracting emission reduction efforts. However, following the 2013 Action Plan on Prevention and Control of Air Pollution, $O_3$ concentrations increased, corresponding with the observed rise in SOC levels discussed in section 3.3.

The correlation between SOC and $O_3$ provides valuable insights into the formation and transformation of ambient SOC. Figure 6 shows the statistical distribution of SOC concentrations across different $O_3$ intervals during various seasons. Results indicate a concurrent increase in OC and $O_3$ concentrations when $O_3$ levels exceed 100 µg m⁻³, a phenomenon typically observed during warmer seasons. In particular, during summer, OC and $O_3$ concentrations rise simultaneously when $O_3$ exceeds 50 µg m⁻³. This trend suggests that the strong solar radiation and high temperatures in summer enhance photochemical reactions, making SOC more influenced by $O_3$. Besides the effects of meteorological conditions, this trend is likely driven primarily by the

oxidation of VOCs by hydroxyl radicals, rather than the gas-to-particle partitioning of low-volatility organic compounds. This oxidation process generates both SOC and $O_3$, both of which have relatively long lifetimes (>12 hours).

To investigate the effects of $T$ and RH on the formation of SOA, the dataset was categorized by seasons. Figure 7 depicts the $T$- and RH-dependent distributions of SOC mass concentrations. During spring, summer, and winter, SOC exhibited the highest mass loadings (>4 μg m$^{-3}$) at T > 25 °C, T > 30 °C, and T > 15 °C, respectively, while showing no significant dependence on RH. This pattern suggests that SOC formation in spring and summer is primarily driven by photochemical processes. It is noteworthy that low temperatures do not significantly reduce SOA formation rates during winter (Huang et al., 2014a).

Additionally, processes such as aqueous-phase oxidation and $NO_3$-radical-initiated nocturnal chemistry likely contribute to, or even dominate, SOA and SOC formation during winter (Hallquist et al., 2009; Saffari et al., 2016). Furthermore, high SOC concentrations were occasionally observed at low $T$ and median RH during autumn, indicating a complex source of SOC in Shanghai during this season. This phenomenon may be attributed to regional transport and local emissions, particularly from biomass burning. The increase in biomass burning activities during the harvest season releases volatile organic compounds

(VOCs), which can further enhance SOC formation through photochemical reactions. Additionally, pollutants from surrounding regions may be transported to Shanghai under favorable meteorological conditions, thereby influencing SOC concentrations.

**3.5 Geographic potential sources of SOC**

In 2014, the Air Pollution Action Plan was fully implemented, coinciding with a period of high carbonaceous aerosol loadings

and reliable data. This makes 2014 particularly suitable for investigating the potential influence of transportation on SOC levels. Previously, Chang et al. (2017) analyzed the geographic sources of OC and EC in Shanghai, so the current focus is solely on SOC. Figure 8a shows the seasonal bivariate polar plots of SOC concentrations. During spring and summer, high SOC concentrations are primarily observed from the southwest direction and under low wind speeds (WS less than 4 m s$^{-1}$). This distribution pattern, with SOC concentrations exceeding 3 μg m$^{-3}$ under wind speeds around 2 m s$^{-1}$, indicates that local

sources are the primary contributors to SOC during these seasons. In contrast, during autumn and winter, high SOC concentrations are associated with high wind speeds (WS > 8 m s$^{-1}$) coming from the northwest. This suggests that SOC during these seasons may be influenced by long-range transport of air pollutants from northern China, carried by northwesterly winds from Siberia to the Yangtze River Delta region.

In Fig. 8b, the PSCF analysis reveals the spatial distribution of SOC using SOC-weighted 72-hour backward trajectories (also

see Fig. S14). During spring, SOC mainly originates from the middle and lower reaches of the Yangtze River and the East China Sea, with significant contributions from central Anhui, southern Jiangsu, central Zhejiang, northern Jiangxi, and northern Fujian. In summer, high SOC values in southern Shanghai primarily come from central and southern Anhui, eastern Zhejiang, and Fujian, while in the north, the main sources are eastern Shandong and offshore regions such as the Yellow Sea and East

China Sea. In autumn, the primary sources of SOC in Shanghai are northern and southern Zhejiang and Anhui. During winter, the main source is northern southern Zhejiang, with additional contributions from long-distance transport from the northern North China Plain extending beyond Inner Mongolia, Mongolia, and Russia. Notably, the pollution belt overlaps with the Yangtze River Economic Belt, one of the most densely populated regions in China, which includes major cities like Wuhan, Changsha, Nanchang, Hefei, Nanjing, and Suzhou, along with numerous industrial complexes featuring petrochemical, iron and steel, and chemical processing facilities. Furthermore, Shanghai's proximity to the sea means that SOC concentrations are influenced by ocean currents and typhoons. In summary, the PSCF analysis underscores the significant role of long-range transport in contributing to SOC pollution in Shanghai.

## 4 Conclusions

This study presents long-term measurements and comprehensive analysis of carbonaceous aerosols in $PM_{2.5}$ in Shanghai. We further estimated POC and SOC levels, examining their temporal variations on interannual, monthly, seasonal, and diurnal scales. Through rigorous statistical analysis and correlation studies with meteorological parameters and pollutant concentrations, the origins, formation mechanisms, and spatial distribution patterns of SOC were elucidated.

Annual SOC mass concentrations ranged from 2.2 to 2.9 µg m$^{-3}$, accounting for 28.0% to 41.9% of total OC. Despite the substantial contribution of SOC to OC, the declining trends observed in OC, EC, and $PM_{2.5}$ underscore the pivotal role of emission reduction strategies in curbing carbonaceous aerosols. Notably, stricter regulations on VOCs post-2017 led to a discernible decrease in SOC levels, indicating the effectiveness of emission control measures. Although this study primarily focuses on the period from 2010 to 2016, this trend provides context for the broader improvement in air quality. During summer, SOC concentrations exhibited a strong correlation with ozone levels, particularly at temperatures exceeding 30 °C, highlighting the importance of photochemical processes in SOC formation. Conversely, winter witnessed the highest SOC concentrations, influenced by both local emissions and long-range transport.

Spatial analysis using PSCF revealed that SOC sources are primarily regional, with contributions from neighboring provinces such as Anhui, Jiangsu, and Zhejiang. Furthermore, long-range transport from the Yangtze River Delta region and beyond significantly impacts SOC levels in Shanghai.

In summary, the findings contribute to a deeper understanding of carbonaceous aerosols, specifically SOC, and their implications for air quality management. By examining the complex interplay between emissions, meteorological factors, and regional transport, this study provides valuable insights for informing future air pollution control strategies in urban environments.

## Data availability

Data related to this article are available upon reasonable request to the corresponding author.

## Author contributions

Yunhua Chang and Jianlin Hu designed the study. Yunhua Chang and Haifeng Yu interpreted the results and wrote the paper with contributions and suggestions from all co-authors.

## Competing interests

The contact author has declared that none of the authors has any competing interests.

## Acknowledgements

This work was supported by the National Key R&D Program of China (grant 2019YFA0607202), the National Natural Science Foundation of China (41975166), Jiangsu Natural Science Fund for Excellent Young Scholars (BK20211594), and the Science and Technology Commission of the Shanghai Municipality (20ZR1447800).

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

 **Table 1.** Descriptive statistics of annual and seasonal changes in EC, OC, OC/EC, POC, and SOC in Shanghai from July 10, 2010, to July 31, 2017. Note that statistics for spring 2017 are not provided due to insufficient data.

| Year | Season | Total(N) | OC | EC | OC/EC | POC | SOC |
|------|--------|----------|------|------|-------|------|------|
| 2010 | annual | 4863 | 6.9±5.1 | 2.8±2.3 | 3.3±1.2 | 4.8±4.5 | 2.2±1.9 |
|      | summer | 1025 | 6.4±4.0 | 2.5±2.0 | 3.9±2.3 | 3.8±3.2 | 2.6±2.1 |
|      | autumn | 1827 | 7.1±6.3 | 2.5±2.1 | 3.4±1.9 | 5.1±5.0 | 2.0±1.9 |
|      | winter | 2011 | 7.1±4.2 | 3.3±2.6 | 2.8±1.6 | 4.9±3.9 | 2.2±1.5 |
| 2011 | annual | 7819 | 6.2±4.1 | 2.3±1.7 | 3.3±1.8 | 4.2±3.4 | 1.9±1.8 |
|      | spring | 2125 | 6.7±3.7 | 2.5±1.6 | 3.2±1.4 | 4.8±3.1 | 2.0±1.6 |
|      | summer | 1797 | 6.4±4.6 | 2.2±1.4 | 3.3±1.6 | 4.6±3.3 | 1.8±1.7 |
|      | autumn | 1940 | 5.7±4.8 | 2.1±1.6 | 3.3±1.7 | 4.1±3.9 | 1.6±1.5 |
|      | winter | 1957 | 5.8±3.1 | 2.3±1.9 | 3.6±2.3 | 3.5±2.9 | 2.3±1.3 |
| 2012 | annual | 4107 | 6.3±4.5 | 2.2±1.8 | 3.4±1.7 | 4.2±3.5 | 2.1±2.0 |
|      | spring | 187 | 4.4±2.7 | 1.3±1.0 | 4.7±2.6 | 2.4±1.9 | 2.0±1.6 |
|      | summer | 1250 | 5.2±2.9 | 1.6±1.2 | 3.9±1.9 | 3.2±2.5 | 2.0±1.5 |
|      | autumn | 1512 | 6.2±4.9 | 2.3±1.9 | 2.9±1.2 | 4.7±3.9 | 1.5±1.4 |
|      | winter | 1158 | 7.8±5.1 | 2.7±2.2 | 3.4±1.3 | 4.9±3.7 | 3.0±2.7 |
| 2013 | annual | 5463 | 9.5±6.4 | 2.7±2.1 | 3.9±1.6 | 6.8±5.3 | 2.7±2.6 |
|      | spring | 1010 | 7.5±4.7 | 2.5±1.8 | 3.5±1.3 | 5.5±4.1 | 2.0±1.9 |
|      | summer | 1543 | 9.4±5.0 | 2.3±1.3 | 4.4±1.7 | 6.6±3.7 | 2.8±2.7 |
|      | autumn | 1459 | 8.4±6.3 | 2.3±1.8 | 3.9±1.5 | 5.7±5.1 | 2.7±2.6 |
|      | winter | 1451 | 12.0±7.9 | 3.6±2.7 | 3.9±1.6 | 9.2±6.8 | 2.8±2.7 |
| 2014 | annual | 6404 | 7.9±4.8 | 2.1±1.6 | 4.5±1.9 | 5.1±3.9 | 2.9±2.1 |
|      | spring | 1282 | 8.7±4.5 | 2.2±1.4 | 4.7±2.2 | 5.5±3.6 | 3.2±2.3 |
|      | summer | 1826 | 6.5±2.9 | 1.7±0.9 | 4.6±1.9 | 3.6±2.1 | 2.9±1.9 |
|      | autumn | 2141 | 6.7±3.2 | 1.7±1.1 | 4.5±1.8 | 4.2±2.7 | 2.5±1.6 |
|      | winter | 1155 | 11.7±7.0 | 3.4±2.3 | 3.9±1.5 | 8.7±5.8 | 3.0±2.9 |
| 2015 | annual | 3274 | 7.0±4.3 | 1.7±1.2 | 5.1±2.9 | 4.1±3.2 | 2.9±2.4 |
|      | spring | 458 | 6.4±3.3 | 2.2±1.2 | 3.1±1.5 | 4.4±2.5 | 2.0±1.8 |
|      | summer | 903 | 6.7±3.6 | 1.5±0.9 | 5.4±2.7 | 3.5±2.3 | 3.2±2.5 |
|      | autumn | 753 | 4.8±2.5 | 1.2±0.8 | 4.9±2.9 | 2.4±1.8 | 2.4±1.6 |
|      | winter | 1160 | 9.0±5.2 | 1.9±1.4 | 5.8±3.2 | 5.5±4.0 | 3.5±2.7 |

|      |        |      |          |         |         |         |         |
|------|--------|------|----------|---------|---------|---------|---------|
|      | annual | 6904 | 6.8±4.0  | 2.2±1.6 | 3.9±3.1 | 4.1±3.1 | 2.7±2.4 |
|      | spring | 1425 | 7.1±3.3  | 1.6±1.1 | 6.3±4.5 | 3.3±2.3 | 3.9±2.2 |
| 2016 | summer | 1747 | 4.8±2.9  | 1.5±0.8 | 3.7±2.0 | 3.3±2.3 | 1.5±1.4 |
|      | autumn | 1782 | 6.2±3.8  | 2.3±1.5 | 3.1±2.1 | 4.2±2.9 | 2.1±2.0 |
|      | winter | 1950 | 8.7±4.6  | 3.3±2.0 | 3.2±2.2 | 5.3±3.8 | 3.3±2.7 |
| 2017 | summer | 674  | 11.8±4.9 | 2.9±1.4 | 4.3±1.6 | 8.2±3.8 | 3.6±2.9 |

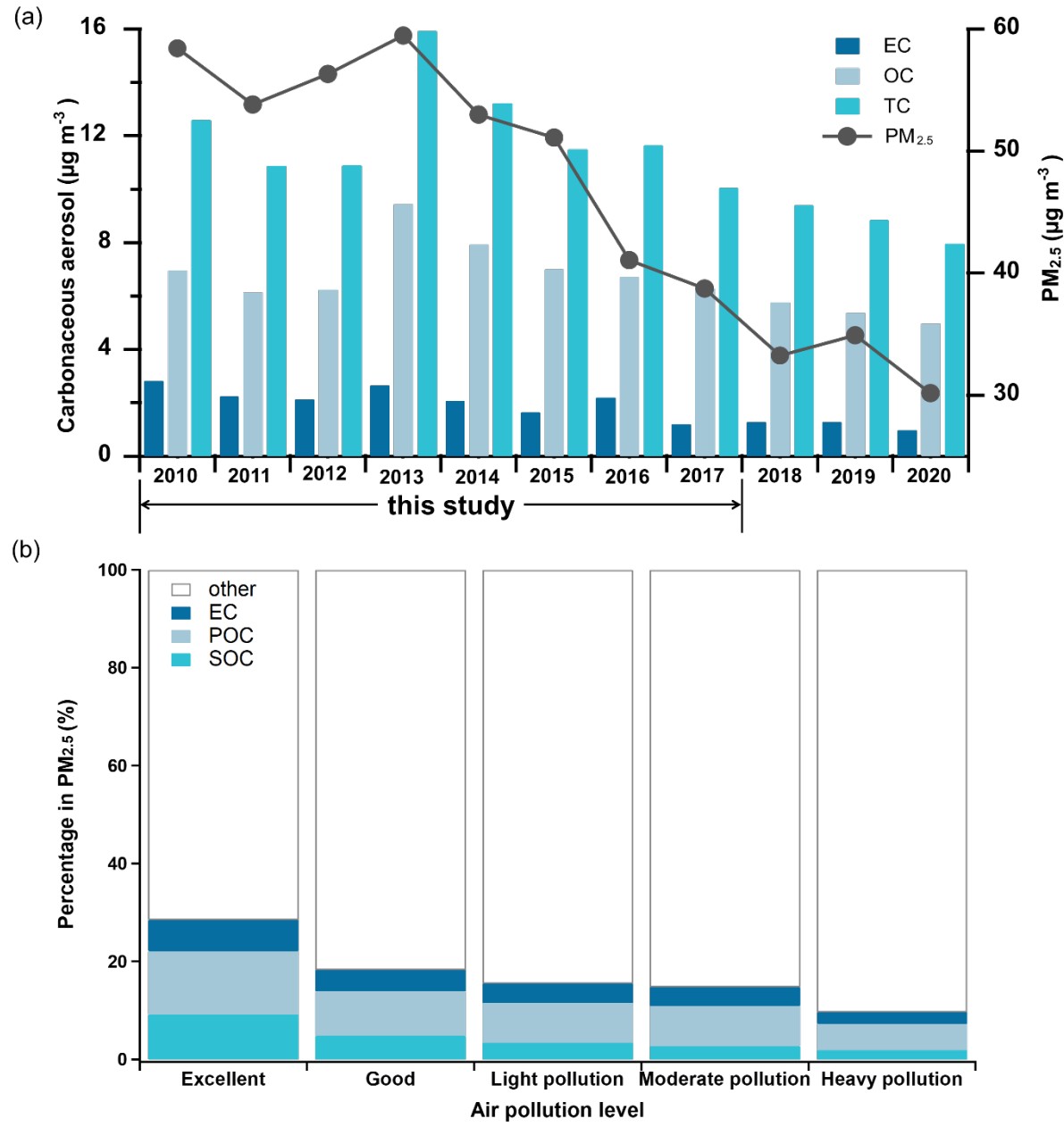

**Figure 1.** Interannual variation of OC, EC, TC and PM$_{2.5}$ from 2010 to 2020 (a) and the proportion of EC, POC and SOC in PM$_{2.5}$ under different air pollution levels (b).


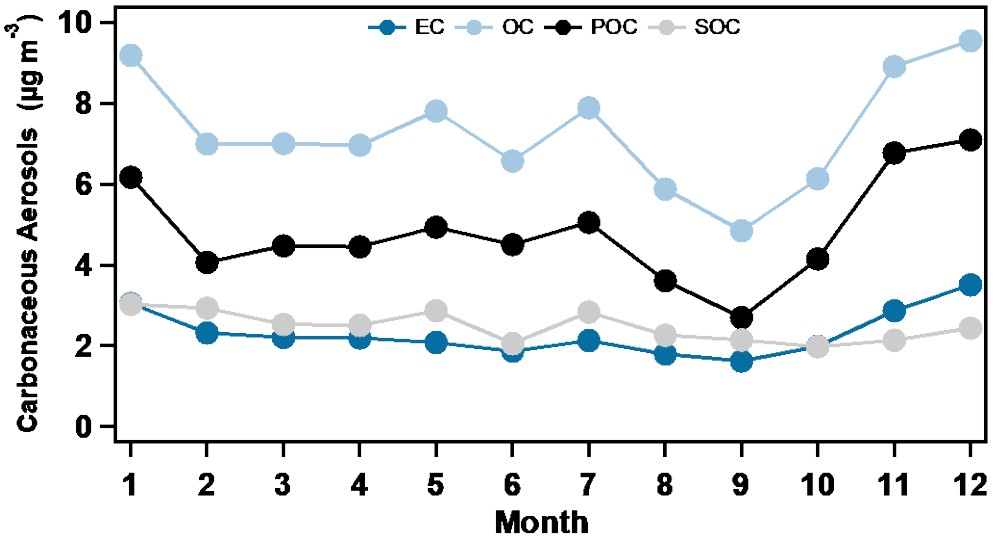

**Figure 2.** Monthly variation of EC, OC, POC, and SOC concentrations during the study period.

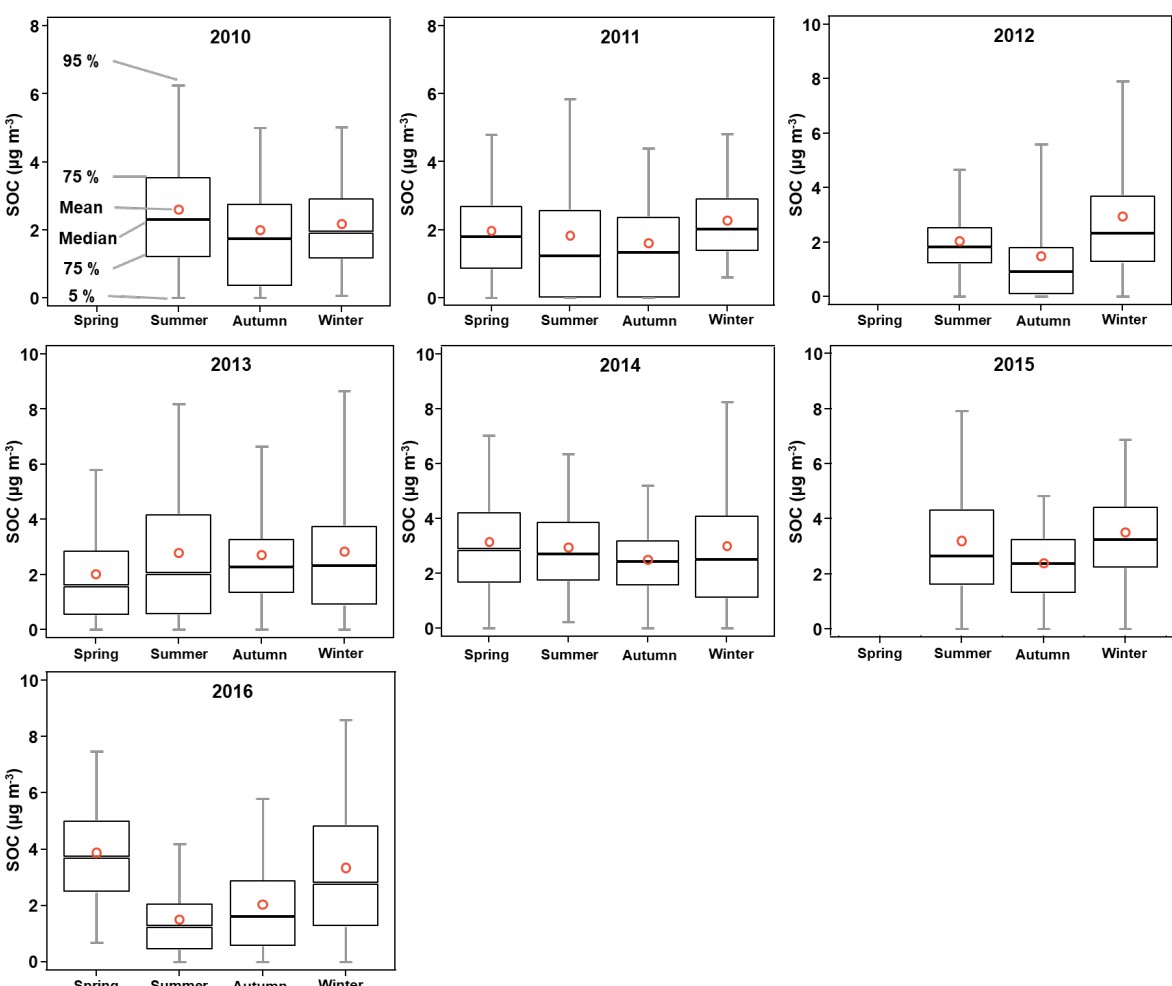

**Figure 3.** SOC concentrations from 2010 to 2016 across different seasons (spring: March, April, May; summer: June, July, August; autumn: September, October, November; winter: December, January, February). For 2017, only summer data is available, hence seasonal changes are not discussed.

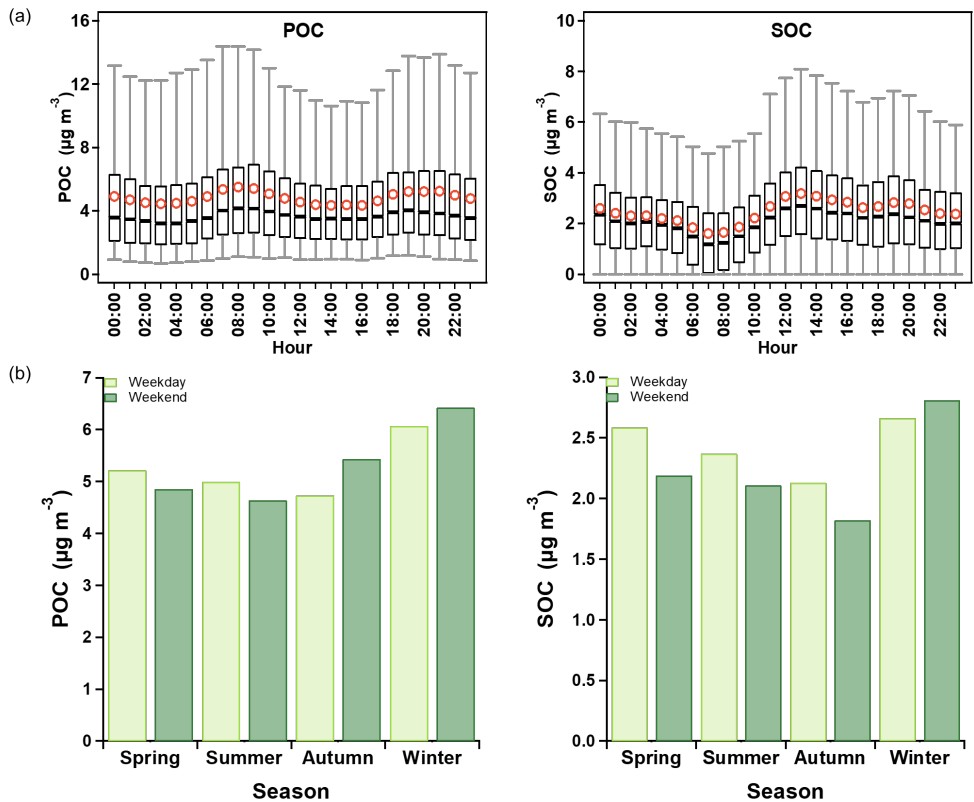

**Figure 4.** Diurnal variation of POC and SOC concentrations (a), and weekend effects on POC and SOC concentrations in different seasons (b).

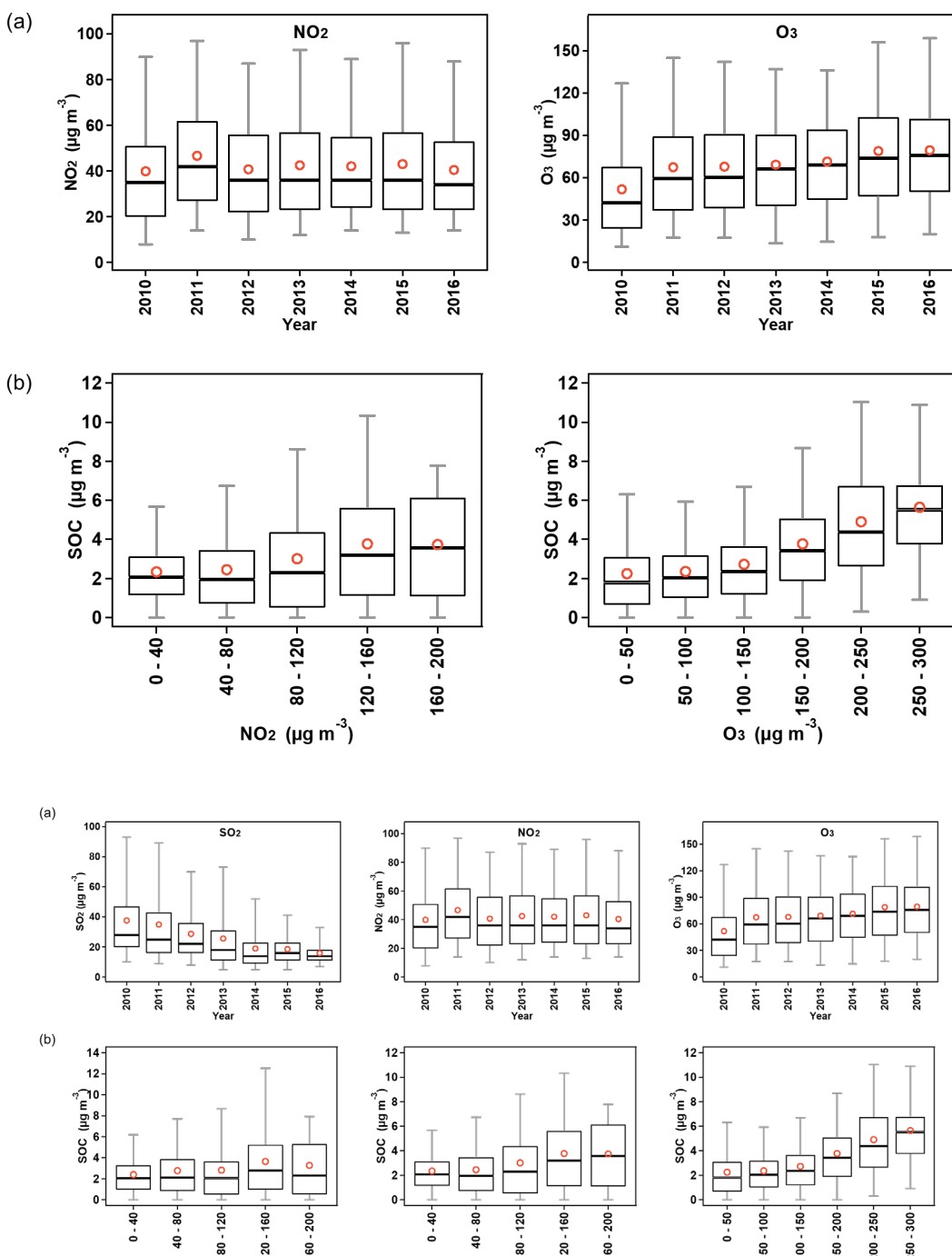

**Figure 5.** Trends in annual average concentrations of $SO_2$, $NO_2$ and $O_3$ from 2010 to 2016 (a), and SOC concentrations under different $NO_2$ and $O_3$ concentration ranges (b).


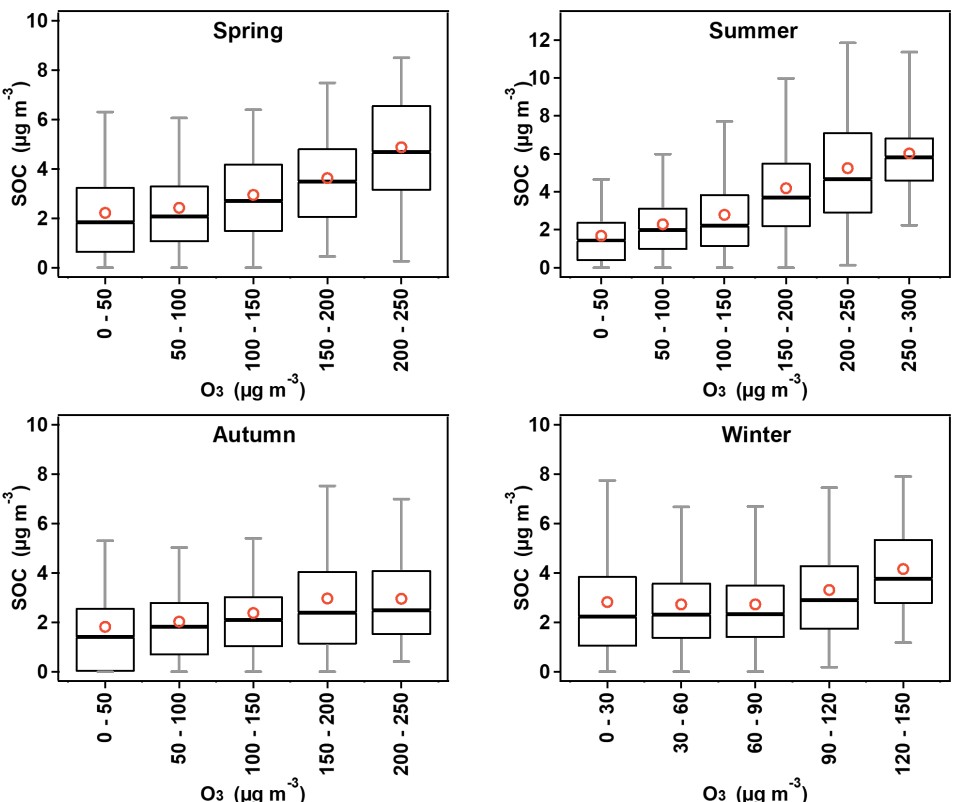


**Figure 6.** SOC concentrations as a function of the $O_3$ concentrations in different seasons.

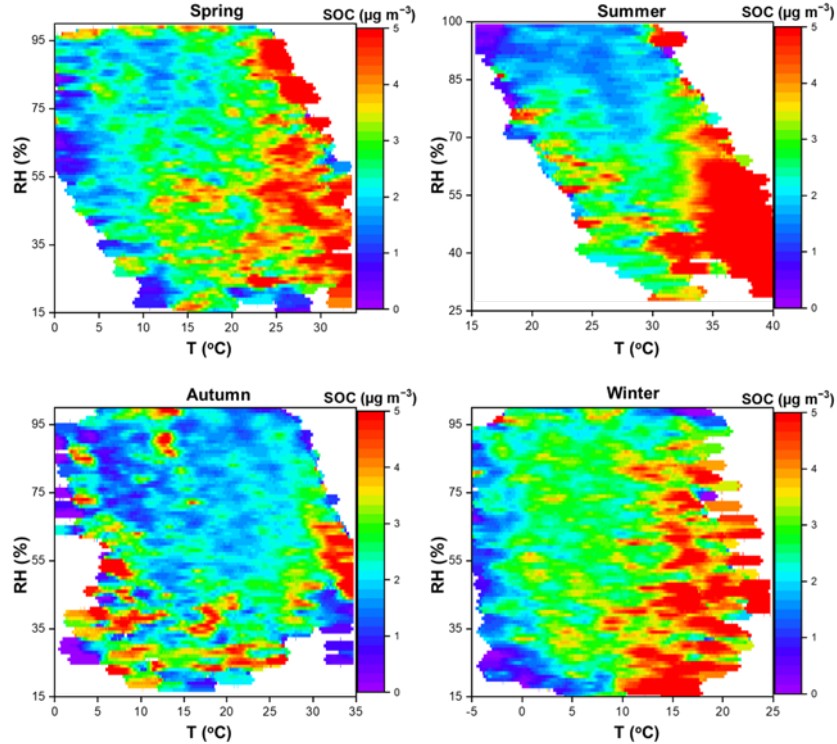

**Figure 7.** RH–*T* dependence of SOC mass concentrations.

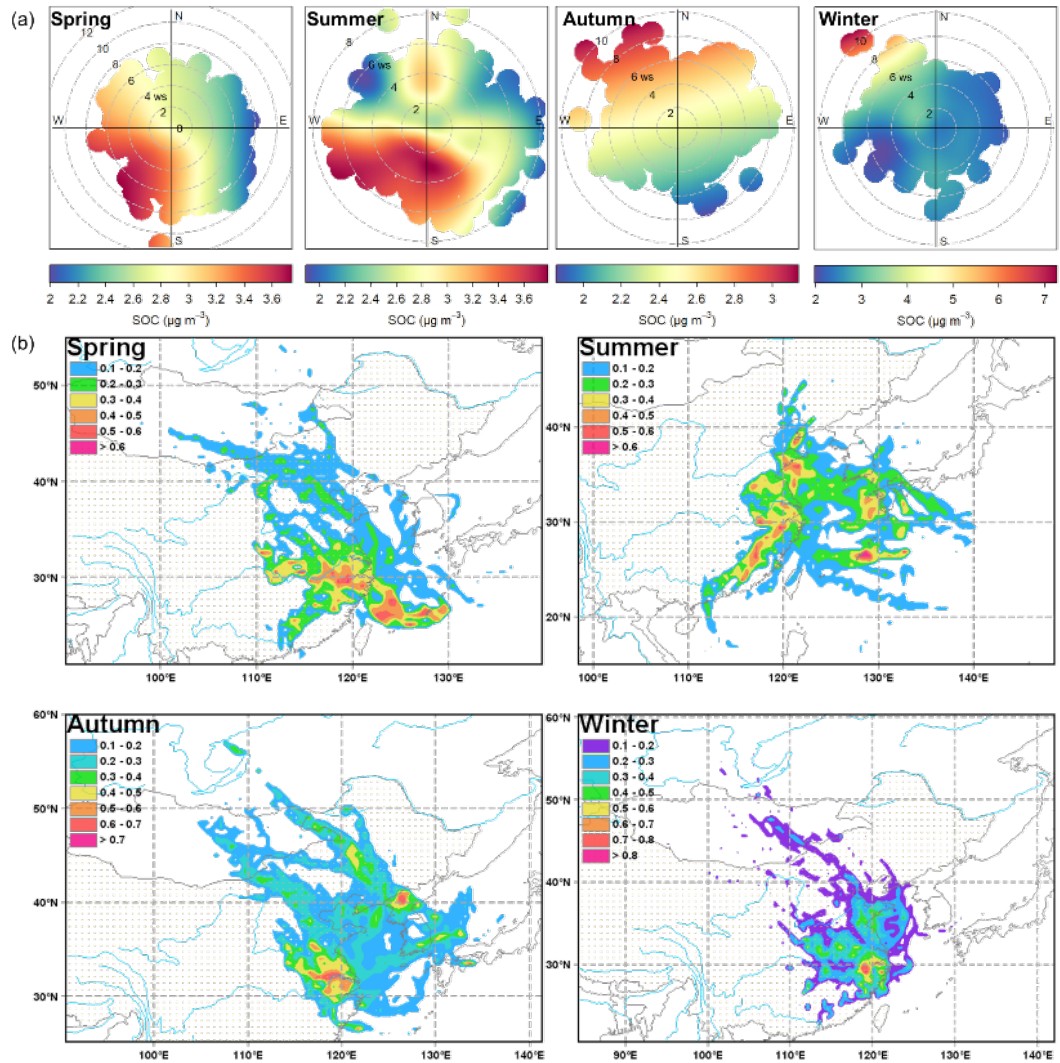

**Figure 8.** (a) Bivariate polar plots of seasonal SOC concentrations (µg m⁻³) in Shanghai from July 2010 to July 2017. SOC concentrations are depicted using a color scale. (b) Potential source contribution function (PSCF) of seasonal SOC in Shanghai in 2014.
