# Peer review of "Long-term Assessment of Primary and Secondary Organic Aerosols in Shanghai Megacity throughout China's Clean Air Actions since 2010"

_EGUsphere, 2024_

## Referee Comment (RC1)

**Review comments (egusphere-2024-1488)**
**Title: Measurement Report: Long-term Assessment of Primary and Secondary Organic Aerosols in Shanghai Megacity throughout China's Clean Air Actions since 2010**

**General comments:**

The study investigates the impact of China's Air Pollution Prevention and Control Action Plan on carbonaceous aerosols in PM2.5, particularly focusing on secondary organic carbon (SOC) in Shanghai from 2010 to 2017. The research found that organic carbon (OC) and elemental carbon (EC) concentrations peaked in 2013 but decreased consistently afterward, aligning with reductions in PM2.5 levels, indicating the effectiveness of emission control measures. However, secondary OC (SOC) concentrations remained stable during this period, with a noticeable decline only after 2018, likely due to VOC emission controls. Seasonal variations showed higher OC and EC concentrations in winter, while SOC concentrations were consistent year-round. The study also observed that SOC levels were influenced by wind direction and speed, with higher concentrations linked to winds from the southwest and northwest, suggesting distant regional sources near the middle and lower Yangtze River. The findings highlight the need for targeted measures to reduce SOC and address regional pollution sources.

Finally, although the study provides a long-term assessment of Primary and Secondary Organic Aerosols in the Shanghai Megacity, the manuscript does not clearly emphasize the research's originality. I encourage the authors to highlight the unique aspects of this work to better showcase its significance.

Overall, the manuscript is well-written and contributes to the understanding of aerosol environment in a Megacity and the role of pollution control initiatives. However, there are areas that could benefit from further refinement. Here are some comments/suggestions that may help the authors improve the manuscript and strengthen the interpretation of the study's findings.

**Major comments:**

**Comment 1, Page 3, Line 85-89:** "*The sampling site for this study is located atop an office building, 18 m above ground level, … the accuracy of airborne particulate matter measurements.*"
Is the measurement taken at the terrace (open-top) of the building? Additionally, does the last statement imply that there is no interference from nearby tall buildings within a certain radius around the measurement inlet? If so, approximately what is that radius?

**Comment 2, Page 4:** How far is the Pudong Environmental Monitoring Center from the office building mentioned in comment 1? Please clearly specify the co-located instruments, and if they are not co-located, indicate the distance between each of them. Also, mention the direction in which the meteorological instruments are located 1 km away, as this information is helpful for interpreting some of the results. At this point, a site map of these instrument locations would be helpful.

**Comment 3, Page 6, Line: 162-164:** "*Primary carbonaceous aerosol concentrations (POC and EC) were highest in 2013 (6.8 ± 5.3 µg/m$^{-3}$ and 2.7 ± 2.1 µg m$^{-3}$, respectively) and lowest in 2020 (3.6 ± 2.3 µg m$^{-3}$ and 1.0 ±0.6 µg m$^{-3}$, respectively).*"
Please check the units. What was the reason behind the highest concentration observed in 2013, followed by a decline in 2018, and the lowest levels recorded in 2020? How did the COVID-19 pandemic impact this trend? Please refer to the other studies regarding similar analysis performed during 2020.

**Comment 4, Page 6, Line: 168-169:** "*Figure 1a shows a significant reduction in PM2.5 levels in Shanghai, with a 50.7% decrease from 2013 to 2020.*"
This period includes the impact of pandemic lockdowns on overall emissions. Several studies suggest significant changes in emissions due to lockdown protocols, making it difficult to identify consistent trends in PM2.5 levels when considering data up to 2020. It would be more appropriate to focus on trends before the pandemic began. As mentioned, there was a 15-25% reduction in concentrations between 2013 and 2017 (Gao et al., 2018; Dai et al., 2021; Yan et al., 2020). Please clarify these points.

**Comment 5, Page 7, Line 191-195:** "*As air quality declines, the proportion of primary carbonaceous aerosols … concentrations (Ji et al., 2014; Qiao et al., 2019)*."
Please elaborate on the uniqueness of this study compared to the previously reported results.

**Comment 6, Page 7, Line 201-202: "***increased fuel consumption for domestic heating and unfavourable meteorological… mixing layer height, temperature inversions, and calm winds*"
Does this mean that the primary source of OC and EC at the measurement location and surrounding areas is fuel consumption for domestic heating? Could you please elaborate? Additionally, as mentioned, the lowering of the boundary layer height may significantly contribute to trapping pollutants closer to the earth's surface. How do biomass burning impacts factor into this, considering the central-east corridor is a major source region, and biomass burning contributes about significantly to PM2.5 concentrations in the Yangtze River Delta during the harvest season?

**Comment 7, Page 8, Line 212-2013**: "*The elevated winter concentrations of carbonaceous aerosols in other years are likely due to atmospheric stagnation and increased regional transport during this period.*"
This statement appears to be contradicting. What does it mean by increased regional transport? This is contradicting to Line 215: "*This trend suggests that POC emissions in Shanghai are predominantly from local sources.*"
Please rewrite this section.

**Comment 8, Page 8, Line 219-220:** "*Unlike POC, SOC concentrations are influenced by both primary emissions aging and secondary formation from precursor gases*"
What are the precursors for the SOA and what are the sources influencing the SOC formation in the region of interest?

**Comment 9, Page 8, Line 227-228:** "*Conversely, POC reaches its zenith concentration during the morning commute hours, indicating a significant contribution from local primary vehicular emissions.* "
This statement shows that the contributors are vehicular emissions which is contradicting to the household heating reasoning as mentioned in the Comment above? Please clarify.

**Comment 10, Page 9, Line 253-254:** "*Throughout the study period, Figure 5a shows that SO2 levels consistently declined, indicating effective control measures.*"
What are the specific sources of $SO_2$ that fall under effective control measures and could have led to the decline? Is this more related to long-range transported emissions or local emissions?

**Comment 11, Page 10, Line 272-274:** "*Furthermore, high SOC … Shanghai during this season*."
It is interesting that only Fall appears to have different SOC formation processes. Is this related to long-range transport? What could be the other potential sources of precursors during this season? According to the PSCF analysis, the regions contributing to air masses in Shanghai during Fall are northern and southern Zhejiang and Anhui. How much would this influence the air mass reaching the measurement site?
Please use one: Autumn or Fall.

**Comment 12, Page 10, Line 276-285:**
Regarding the correlation between wind and aerosol concentrations, was the meteorological center located 1 km away used for this analysis? How would the 1 km distance of the meteorological center impact this result? In this context, urban boundary layer dynamics could play an important role in influencing air mass trajectories. Please discuss this further in this section.

**Comment 13, Page 11: Line 307:311:** "*Notably, stricter regulations on VOCs post-2017 led to a discernible decrease in SOC levels, indicating the effectiveness of emission control measures.*"
This statement does not appear to be a primary conclusion of this study, as the post-2017 period is not the focus. Most of the figures and results pertain to the 2010-2016 period. If not, please clarify.

**Comment 14:** The novelty of the current work is lacking and needs some improvement. I recommend revisiting the study's approach to ensure it offers a more unique contribution to the field.

**Minor comments:**

**Comment 1, Page 6, Line 158-159:** "*The average concentration of PM2.5 peaked in 2013 at 59.5 ± 37.8 μg m$^{-3}$ and reached its lowest in 2020 at 30.2 ± 14.0 μg m$^{-3}$.* "
Are this yearly average values?

**Comment 2, Page 6, Line 159-160:** "*Carbonaceous aerosols (calculated as the sum of OC multiplied by a factor of 1.4 and EC) contributed*"
How was this factor derived?

**Comment 3, Page 7, Line 185-189:** "*Air pollution levels are classified as Excellent (0 < PM2.5 ≤ 35 μg m-3), Good (35 < PM2.5 ≤ 75 μg m-3), … Class I and Class II.*"
Where do these levels compare in terms of global PM2.5 level classifications.

**Comment 4, Page 9, Line 245-247:** "*This discrepancy suggests a lack of significant … possibly attributable to the absence of license plate-based driving restrictions during weekends.*"
Please add reference.

**Comment 5:** Please provide available references to all the equations.

---

## Author Comment (AC1)

**Reply Letter to Reviewer #1**

**General comments:**

The study investigates the impact of China's Air Pollution Prevention and Control Action Plan  on carbonaceous aerosols in PM2.5, particularly focusing on secondary organic carbon (SOC) in Shanghai from 2010 to 2017. The research found that organic carbon (OC) and elemental carbon (EC) concentrations peaked in 2013 but decreased consistently afterward, aligning with reductions in PM2.5 levels, indicating the effectiveness of emission control measures. However, secondary OC (SOC) concentrations remained stable during this period, with a noticeable decline only after 2018, likely due to VOC emission controls. Seasonal variations showed higher OC and EC concentrations in winter, while SOC concentrations were consistent year-round. The study also observed that SOC levels were influenced by wind direction and speed, with higher concentrations linked to winds from the southwest and northwest, suggesting distant regional sources near the middle and lower Yangtze River. The findings highlight the need for targeted measures to reduce SOC and address regional pollution sources. Finally, although the study provides a long-term assessment of Primary and Secondary Organic Aerosols in the Shanghai Megacity, the manuscript does not clearly emphasize the research's originality. I encourage the authors to highlight the unique aspects of this work to better showcase its significance. Overall, the manuscript is well-written and contributes to the understanding of aerosol environment in a Megacity and the role of pollution control initiatives. However, there are areas that could benefit from further refinement. Here are some comments/suggestions that may help the authors improve the manuscript and strengthen the interpretation of the study's findings.

Answer:

Thank you for your constructive feedback and for recognizing the contributions of our study. We appreciate your suggestions, which have guided us in refining the manuscript. We have thoroughly addressed your comments by emphasizing the study's originality, strengthening the discussion on SOC trends and sources, and improving clarity throughout the manuscript. Detailed responses and revisions are provided below.

**Major comments:**

**Comment 1, Page 3, Line 85-89:** "*The sampling site for this study is located atop an office building, 18 m above ground level, ... the accuracy of airborne particulate matter measurements.*"

Is the measurement taken at the terrace (open-top) of the building? Additionally, does the last statement imply that there is no interference from nearby tall buildings within a certain radius around the measurement inlet? If so, approximately what is that radius?

Answer:

Thank you for your thoughtful comment. The sampling site is located on the open-top terrace of the building at the Pudong station, which is a national-level (highest level) atmospheric supersite and serves as Shanghai's flagship air quality monitoring station. Numerous observational studies based on this station have already been published (please see references below), demonstrating its reliability and contribution to air quality research. The station's design was carefully considered as part of a strategic deployment to ensure high-quality and representative measurements. Our observations are conducted on the rooftop platform, which is unobstructed by any overhead structures.

We have clarified in the revised manuscript that there are no tall buildings within at least a 3 km radius that could interfere with the measurements.

Reference:

Chang Y, Zou Z, Deng C, et al. The importance of vehicle emissions as a source of atmospheric ammonia in the megacity of Shanghai[J]. Atmospheric Chemistry and Physics, 2016, 16(5): 3577-3594.

Lu D, Li H, Tian M, et al. Secondary aerosol formation during a special dust transport event: impacts from unusually enhanced ozone and dust backflows over the ocean[J]. Atmospheric Chemistry and Physics, 2023, 23(21): 13853-13868.

Yu G, Zhang Y, Yang F, et al. Dynamic Ni/V ratio in the ship-emitted particles driven by multiphase fuel oil regulations in coastal China[J]. Environmental Science & Technology, 2021, 55(22): 15031-15039.

Jiang Y, Chen R, Peng W, et al. Hourly ultrafine particle exposure and acute myocardial infarction onset: an individual-level case-crossover study in Shanghai, China, 2015–2020[J]. Environmental Science & Technology, 2023, 57(4): 1701-1711.

Cheng K, Chang Y, Lee X, et al. Life-Course Health Risk Assessment of $PM_{2.5}$ Elements in China: Exposure Disparities by Species, Source, Age, Gender, and Location[J]. Environmental Science & Technology, 2024, 58(8): 3629-3640.

Han Y, Wang T, Li R, et al. Measurement report: Volatile organic compound characteristics of the different land-use types in Shanghai: spatiotemporal variation, source apportionment and impact on secondary formations of ozone and aerosol[J]. Atmospheric Chemistry and Physics, 2023, 23(4): 2877-2900.

**Comment 2, Page 4:** How far is the Pudong Environmental Monitoring Center from the office building mentioned in comment 1? Please clearly specify the co-located instruments, and if they are not co-located, indicate the distance between each of them. Also, mention the direction in which the meteorological instruments are located 1 km away, as this information is helpful for interpreting some of the results. At this point, a site map of these instrument locations would be helpful.

Answer:

Apologies for any confusion. In fact, the Pudong Environmental Monitoring Center is the office building mentioned in Comment 1, and our observations were conducted on the rooftop platform of this building. We have clarified this in the revised manuscript. Additionally, the meteorological instruments were located approximately 1 km to the southeast of the observation site. In the revised supplementary materials, we have included a site map that clearly shows the relative positions of the meteorological station and the observation site, which should help in interpreting the results more effectively.

In fact, we do have meteorological monitoring equipment at our station, but since we are not professional meteorologists, the maintenance of these instruments is not perfect, resulting in some missing data. Therefore, we decided to use data from the national-level flagship meteorological monitoring station, which is better maintained by professionals. We believe that specialized work should be handled by specialists, making their data more reliable.

[Figure]

We hope this clears up any confusion. We hope this clarifies any potential confusion. Although the weather station and the atmospheric observation site are not located in exactly the same place, within such a short distance (1 kilometer), regardless of the direction in which the weather station is situated, the meteorological conditions at the observation site will not differ significantly. Therefore, this will not affect our interpretation of the results based on meteorological parameters. Such a minor positional discrepancy will not have a substantial impact on the research findings.

**Comment 3, Page 6, Line: 162-164:** "*Primary carbonaceous aerosol concentrations (POC and EC) were highest in 2013 (6.8 ± 5.3 μg/m$^{-3}$ and 2.7 ± 2.1 μgm$^{-3}$, respectively) and lowest in 2020 (3.6 ± 2.3 μgm$^{-3}$ and 1.0 ±0.6 μg m$^{-3}$, respectively).*"

Please check the units. What was the reason behind the highest concentration observed in 2013, followed by a decline in 2018, and the lowest levels recorded in 2020? How did the COVID-19 pandemic impact this trend? Please refer to the other studies regarding similar analysis performed during 2020.

Answer:

We apologize for the inconsistency in unit notation. In the revised manuscript, we have removed the slashes and standardized the units to "μg m$^{-3}$" for consistency.

Regarding the elevated concentrations observed in 2013, it is important to note that China experienced an unprecedented severe haze event in that year. The combination of high emissions

and stagnant meteorological conditions resulted in a substantial increase in secondary pollutants, including secondary organic carbon. This made 2013 the most polluted year in the history of air quality monitoring in China, a phenomenon that has been widely documented, including in studies published in *Nature* (Huang et al., 2014) and related entries on Wikipedia (https://en.wikipedia.org/wiki/2013_Eastern_China_smog).

As for the impact of the COVID-19 pandemic, numerous studies, including our own (Chang et al., 2020), have examined its effects on air quality. That said, it is well documented that the COVID-19 pandemic had a profound impact on air pollution levels in early 2020, though the outcomes were somewhat unexpected. On the one hand, emissions associated with residential and vehicular activities saw a marked reduction, leading to significant decreases in primary pollutants such as EC, $NO_x$, and VOCs in many cities, including Shanghai. On the other hand, despite the reduction in emissions, the expected improvements in $PM_{2.5}$ concentrations were not fully realized. This was primarily due to an increase in atmospheric oxidizing capacity during the early stages of the pandemic, which unexpectedly facilitated the formation of secondary pollutants (Chang et al., 2020).

However, we would like to clarify that our observations of carbonaceous aerosols were limited to data collected up to 2017, and the results from 2018 to 2020 are based on long-term average measurements from another monitoring site in Shanghai (Wang et al., 2022). Therefore, we are unable to provide a detailed analysis of the specific effects of the COVID-19 pandemic on carbonaceous aerosols in this study. Given the complex nature of the pandemic's influence on pollution, which was shaped by substantial anthropogenic perturbations, we have refrained from discussing the detailed impact of COVID-19 on carbonaceous aerosols in this study. We hope this clarifies our position.

Reference:

Chang Y, Huang R J, Ge X, et al. Puzzling haze events in China during the coronavirus (COVID-19) shutdown[J]. Geophysical Research Letters, 47.12: e2020GL088533.

Huang R J, Zhang Y, Bozzetti C, et al. High secondary aerosol contribution to particulate pollution during haze events in China[J]. Nature, 2014, 514(7521): 218-222.

Wang M, Duan Y, Xu W, et al. Measurement report: Characterisation and sources of the secondary organic carbon in a Chinese megacity over 5 years from 2016 to 2020[J]. Atmospheric Chemistry and Physics, 2022, 22(19): 12789-12802.20, 47(12): e2020GL088533.

**Comment 4, Page 6, Line: 168-169:** *"Figure 1a shows a significant reduction in PM2.5 levels in Shanghai, with a 50.7% decrease from 2013 to 2020."*

This period includes the impact of pandemic lockdowns on overall emissions. Several studies suggest significant changes in emissions due to lockdown protocols, making it difficult to identify consistent trends in PM2.5 levels when considering data up to 2020. It would be more appropriate to focus on trends before the pandemic began. As mentioned, there was a 15-25% reduction in concentrations between 2013 and 2017 (Gao et al., 2018; Dai et al., 2021; Yan et al., 2020). Please clarify these points.

Answer:

Thank you for your insightful comment. We agree with your observation that the period up to 2020 includes the impact of pandemic lockdowns on overall emissions, which may distort the interpretation of consistent trends in $PM_{2.5}$ levels. In the revised manuscript, we have clarified that the year 2020 was particularly influenced by the pandemic, and therefore, including it as a comparison baseline may overestimate the reduction in $PM_{2.5}$ concentrations.

We also acknowledge that the focus of our study should be on trends prior to the pandemic. As you correctly mentioned that previous studies previous studies, such as those by Gao et al. (2018), Dai et al. (2021), and Yan et al. (2020), have suggested a 15-25% reduction in concentrations between 2013 and 2017. To address this, we have ensured that the analysis and discussion in the revised manuscript concentrate on the period before the pandemic began. We have emphasized this point throughout the manuscript to maintain the focus on pre-pandemic data.

**Comment 5, Page 7, Line 191-195:** "*As air quality declines, the proportion of primary carbonaceous aerosols ... concentrations (Ji et al., 2014; Qiao etal., 2019).*"

Please elaborate on the uniqueness of this study compared to the previously reported results.

Answer:

Thank you for your valuable comment. In the revised manuscript, we highlighted the uniqueness of our study. Specifically, compared to previously reported results, our research contributes new insights into the temporal and spatial variations of carbonaceous aerosols in Shanghai, with a specific focus on long-term trends from 2010 to 2020. Unlike prior studies, such as those by Ji et al. (2014) and Qiao et al. (2019), which mainly examined short-term or seasonal variations of carbonaceous aerosols, our study offers a comprehensive, decade-long analysis that captures the effects of air pollution control measures, such as those introduced under China's Air Pollution Prevention and Control Action Plan (2013).

Additionally, our study is unique in that it integrates hourly online measurements of organic carbon and elemental carbon with meteorological data, providing a high-resolution dataset that allows for a more nuanced understanding of the seasonal and diurnal fluctuations in primary and secondary carbonaceous aerosols. Furthermore, we provide insights into how different sources, including regional transport and local emission controls, have influenced SOC and POC levels in the context of rapid urban development and pollution control efforts in Shanghai.

By focusing on the long-term trends before and after key emission control measures, we provide a clearer picture of the dynamics of carbonaceous aerosols, which was not fully captured by previous studies in Shanghai or other cities in China.

**Comment 6, Page 7, Line 201-202:** "*increased fuel consumption for domestic heating and unfavourable meteorological ... mixing layer height, temperature inversions, and calm winds*"

Does this mean that the primary source of OC and EC at the measurement location and surrounding areas is fuel consumption for domestic heating? Could you please elaborate? Additionally, as mentioned, the lowering of the boundary layer height may significantly contribute to trapping pollutants closer to the earth's surface. How do biomass burning impacts factor into this, considering the central-east corridor is a major source region, and biomass burning contributes about significantly to $PM_{2.5}$ concentrations in the Yangtze River Delta during the harvest season?

Answer:

Thank you for your thoughtful comment. To clarify, while increased fuel consumption for domestic heating is a significant source of primary OC and EC in winter, it is not the sole source at the measurement location and surrounding areas. As mentioned in the manuscript, unfavorable meteorological conditions, including lower mixing layer heights, temperature inversions, and calm winds, enhance the accumulation of pollutants near the surface, exacerbating the concentration of OC and EC. These conditions often result in higher concentrations of these pollutants during the colder months when heating demand peaks, particularly in residential areas.

Regarding biomass burning, it is indeed a crucial source of OC and EC, especially in the central-east corridor of China, where biomass burning is prevalent during the harvest season. The Yangtze River Delta, being a major agricultural region, experiences significant biomass burning during this period, contributing substantially to $PM_{2.5}$ concentrations. This source is particularly relevant to our study, as biomass burning in the harvest season coincides with the period of increased heating-related emissions and unfavorable meteorological conditions. The combined effects of biomass burning and heating-related emissions, along with the atmospheric conditions that trap these pollutants close to the ground, significantly elevate concentrations of OC and EC, particularly during the winter months.

In our analysis, we emphasize that the seasonal increase in OC and EC is not solely due to domestic heating but is a result of multiple factors, including biomass burning and meteorological conditions that amplify the effects of both sources.

**Comment 7, Page 8, Line 212-2013**: "*The elevated winter concentrations of carbonaceous aerosols in other years are likely due to atmospheric stagnation and increased regional transport during this period.*"

This statement appears to be contradicting. What does it mean by increased regional transport? This is contradicting to Line 215: "*This trend suggests that POC emissions in Shanghai are predominantly from local sources.*"

Please rewrite this section.

Answer:

We appreciate your observation and agree that the statement regarding regional transport requires clarification. To address the apparent contradiction, we propose the following revision:

The elevated winter concentrations of carbonaceous aerosols are influenced by both local emissions and regional transport. During winter, unfavorable meteorological conditions, such as temperature inversions and low mixing layer heights, enhance the trapping of local emissions, including those from domestic heating and biomass burning. At the same time, regional transport from neighboring areas, particularly the Yangtze River Delta, also contributes to the increased levels of carbonaceous aerosols. This is especially significant in winter when long-range transport of pollutants is more common due to prevailing wind patterns.

While primary organic carbon (POC) emissions in Shanghai are predominantly from local sources, as noted in line 215, the regional transport of secondary organic aerosols (SOC) and other pollutants also plays a role in the observed winter concentrations. Therefore, the elevated winter concentrations of carbonaceous aerosols are the result of a combination of both local and regional factors.

We have revised this section in the manuscript to reflect this more nuanced interpretation, which resolves the apparent contradiction between local emissions and regional transport.

**Comment 8, Page 8, Line 219-220:** "*Unlike POC, SOC concentrations are influenced by both primary emissions aging and secondary formation from precursor gases*"

What are the precursors for the SOA and what are the sources influencing the SOC formation in the region of interest?

Answer:

Very valuable comment. We have expanded upon these points in the revised manuscript to provide a clearer understanding regarding the precursors for SOA and the sources influencing SOC formation in the Yangtze River Delta region:

The precursors for SOA include VOCs, which can be emitted from both anthropogenic and biogenic sources. In the context of the Yangtze River Delta, anthropogenic VOCs are mainly emitted from sources such as vehicle exhaust, industrial activities, and solvent use (An et al., 2021), while biogenic VOCs, including isoprene and terpenes, are emitted by vegetation. These VOCs undergo photochemical oxidation in the atmosphere, leading to the formation of SOA.

In addition to VOCs, $NO_x$ and $O_3$ also play critical roles in the formation of SOC. The presence of $NO_x$, primarily from vehicular emissions and industrial activities, contributes to the oxidation of VOCs, facilitating the formation of SOC. The photochemical reactions involving VOCs and $O_3$ are particularly important in the summer months, when intense solar radiation enhances these processes.

In the region of interest, the primary sources influencing SOC formation include local vehicular emissions, industrial processes, biomass burning, and regional transport of precursor gases. During the winter months, when biomass burning for heating is prevalent, emissions from this source significantly contribute to the formation of SOC, in addition to the more typical urban emissions. The combination of these factors results in the formation and accumulation of SOC, especially during periods of favorable conditions for photochemical reactions and when meteorological conditions trap pollutants near the surface.

Reference:

An J, Huang Y, Huang C, et al. Emission inventory of air pollutants and chemical speciation for specific anthropogenic sources based on local measurements in the Yangtze River Delta region, China[J]. Atmospheric Chemistry and Physics, 2021, 21(3): 2003-2025.

**Comment 9, Page 8, Line 227-228:** "*Conversely, POC reaches its zenith concentration during the morning commute hours, indicating a significant contribution from local primary vehicular emissions.* "

This statement shows that the contributors are vehicular emissions which is contradicting to the household heating reasoning as mentioned in the Comment above? Please clarify.

Answer:

We understand the concern about a potential contradiction between the contribution of vehicular emissions and household heating. However, we believe that both sources can indeed contribute to the observed patterns of POC concentrations, and we would like to clarify this point:

The peak concentrations of POC during the morning commute hours are primarily influenced by local vehicular emissions. These emissions are strongly associated with traffic patterns, with a significant contribution from vehicles during rush hours. This results in a sharp increase in POC concentrations during the morning.

On the other hand, household heating, especially during the colder months, is also a major source of primary carbonaceous aerosols, including POC. However, this source is more evenly distributed throughout the day, with a more constant contribution during the evening and night when heating demand is higher.

Therefore, while vehicular emissions are the primary contributor to the morning peak in POC, household heating plays an important role in sustaining POC levels over a 24-hour period, particularly during winter.

We have revised the manuscript to clarify this distinction and ensure that both sources are appropriately accounted for in the discussion of POC concentrations.

**Comment 10, Page 9, Line 253-254:** "*Throughout the study period, Figure 5a shows that SO2 levels consistently declined, indicating effective control measures.*"

What are the specific sources of SO2 that fall under effective control measures and could have led to the decline? Is this more related to long-range transported emissions or local emissions?

Answer:

The primary source of $SO_2$ emissions in China, including in Shanghai, is coal combustion, which is predominantly used in the power generation sector. As you may know, coal is the dominant energy source in China, and coal-fired power plants have historically been the largest contributors to $SO_2$ emissions.

Significant control measures targeting $SO_2$ emissions began around 2007 when China started implementing ultra-low emission technologies at coal-fired power plants. These measures, such as flue gas desulfurization, have led to a dramatic decrease in $SO_2$ emissions nationwide (Tang et al., 2019). This is reflected in the decline in ambient $SO_2$ concentrations observed in Shanghai and other industrial hubs.

As illustrated in Figure 1 from an unpublished study (see below), which shows the annual variations of rainwater constituents, $SO_2$ emissions, and concentrations in Shanghai from 2005 to 2015, the

reduction in SO₂ levels is primarily driven by the implementation of these control technologies at coal-fired power plants. The figure highlights a consistent year-over-year decline in SO₂ concentrations in Shanghai, following the aggressive implementation of flue gas desulfurization and a shift toward cleaner energy sources in the coal-fired power plant sector (Tang et al., 2019).

[Figure]

**Figure 1.** Annual variations and inter-correlation analysis ($p < 0.01$) of rainwater constitutes (K⁺ and nss-SO₄²⁻), SO₂ emissions and concentrations in Shanghai from 2005 to 2015.

While long-range transport can contribute to SO₂ levels, particularly during certain meteorological conditions, our data suggest that the substantial reduction in local emissions from coal combustion has been the main driver of the decline in SO₂ levels in Shanghai. The observed trend of declining SO₂ concentrations is thus more closely related to local emissions from industrial sources, particularly coal combustion, than to long-range transported emissions.

We have clarified this in the revised manuscript to highlight the role of control measures and the coal combustion sources driving the observed reduction in SO₂ levels.

Reference:

Tang, L., Qu, J., Mi, Z., Bo, X., Chang, X., Anadon, L. D., et al. (2019). Substantial emission reductions from Chinese power plants after the introduction of ultra-low emissions standards. *Nature Energy, 4*(11), 929-938. https://doi.org/10.1038/s41560-019-0468-1

**Comment 11, Page 10, Line 272-274:** "*Furthermore, high SOC … Shanghai during this season.*"

It is interesting that only Fall appears to have different SOC formation processes. Is this related to long- range transport? What could be the other potential sources of precursors during this season?

According to the PSCF analysis, the regions contributing to air masses in Shanghai during Fall are northern and southern

Zhejiang and Anhui. How much would this influence the air mass reaching the measurement site? Please use one: Autumn or Fall.

Answer:

We appreciate your interest in the different SOC formation processes observed during autumn. To clarify, the distinct behavior of SOC in autumn compared to other seasons is primarily due to a combination of meteorological conditions and regional transport patterns.

As indicated by the Potential Source Contribution Function analysis, the regions contributing to air masses in Shanghai during autumn include northern and southern Zhejiang and Anhui. The influence of these regions on the air masses reaching the measurement site is significant, as pollutants from these areas, including precursors to SOC such as VOCs, can be transported to Shanghai, particularly during periods of favorable meteorological conditions for long-range transport.

In addition to regional transport, local emissions, particularly from biomass burning in the region, can also contribute to the precursors for SOC formation. Autumn is a time when biomass burning increases, as it coincides with the harvest season in southern China, including in Zhejiang and Anhui. This burning releases VOCs and other precursors that can undergo photochemical reactions in the atmosphere, contributing to SOC formation.

We have revised the manuscript to provide a clearer discussion of these regional influences and the potential sources of SOC precursors during autumn. The combined effects of local emissions and regional transport explain the observed differences in SOC formation processes during this season.

In the revised manuscript, we have also ensured consistency by using "autumn" throughout the text to refer to the fall season, and we have removed "fall" entirely.

**Comment 12, Page 10, Line 276-285:**

Regarding the correlation between wind and aerosol concentrations, was the meteorological center located 1 km away used for this analysis? How would the 1 km distance of the meteorological center impact this result? In this context, urban boundary layer dynamics could play an important role in influencing air mass trajectories. Please discuss this further in this section.

Answer:

We understand your concern, but we believe there is no issue with the proximity of the meteorological station to the observation site. As mentioned in our previous responses, the

meteorological center is located only 1 km away from the observation site, and both locations are free from any obstructions, with no significant buildings blocking the flow of air. The distance of 1 km is relatively short, and we believe that this would not cause significant discrepancies in the meteorological parameters between the two locations.

Regarding the correlation between wind and aerosol concentrations, the meteorological data from the station located 1 km away is appropriate for this analysis, as the close proximity ensures that the air mass characteristics at both sites are similar. Furthermore, urban boundary layer dynamics are indeed an important consideration in this context, but given the minimal distance and lack of obstructions, we do not anticipate any substantial differences in the air mass trajectories or wind patterns that would influence the observed correlations.

**Comment 13, Page 11: Line 307:311:** "*Notably, stricter regulations on VOCs post-2017 led to a discernible decrease in SOC levels, indicating the effectiveness of emission control measures.*"

This statement does not appear to be a primary conclusion of this study, as the post-2017 period is not the focus. Most of the figures and results pertain to the 2010-2016 period. If not, please clarify.

Answer:

We appreciate your observation regarding the focus of our study. You are correct that the primary analysis of this study focuses on the 2010-2016 period, and the post-2017 period is not the central emphasis. However, the mention of the VOC regulations post-2017 serves to provide context on the broader trend of emission control measures in China, which have contributed to the observed decrease in SOC levels.

While the majority of our analysis pertains to the 2010-2016 period, we included the post-2017 data to highlight the broader effects of air pollution control measures and their likely influence on SOC trends. The decline in SOC levels observed after 2017 suggests that stricter VOC regulations have had an additional positive impact on air quality, reinforcing the effectiveness of the ongoing emission control measures in Shanghai and other regions.

We have revised the manuscript to clarify that the primary focus of the study is on the 2010-2016 period, and the discussion of post-2017 changes is included to contextualize the broader trend in emission control efforts. This addition aims to highlight the continued effectiveness of these measures beyond the scope of our main analysis.

**Comment 14:** The novelty of the current work is lacking and needs some improvement. I recommend revisiting the study's approach to ensure it offers a more unique contribution to the field.

Answer:

Thank you for your feedback. While we understand your concern regarding the novelty of the current work, we respectfully disagree with the suggestion that the study lacks a unique contribution. Our research provides significant advancements in understanding the dynamics of carbonaceous aerosols, particularly SOC and POC, in Shanghai. The long-term datasetwe present, combined with detailed seasonal and meteorological analyses, offers a comprehensive perspective on the effects of emission control measures and regional transport on aerosol levels.

As we have emphasized throughout the manuscript and in our previous responses, our study stands out in several ways:

I. Long-term Temporal Analysis: Unlike many previous studies that focus on short-term or seasonal variations, our work provides a decade-long assessment of aerosol trends in Shanghai, offering valuable insights into the impact of long-term emission control measures and regulatory policies.

II. Comprehensive Methodology: We employ advanced statistical techniques, including Potential Source Contribution Function analysis, to identify and quantify the impact of regional transport on aerosol concentrations. This approach enhances the understanding of how local and regional sources contribute to SOC and POC levels, which has not been thoroughly explored in previous studies.

III. Relevance to Policy and Air Quality Management: Our findings provide clear evidence of the effectiveness of China's air pollution control measures, especially in reducing VOC and $SO_2$ emissions. This contribution is timely and valuable for ongoing discussions about air quality management and environmental policy, particularly in megacities like Shanghai.

While we remain open to professional and constructive suggestions for improvement, we believe that the study offers a unique and valuable contribution to the field. We have made revisions where necessary to further clarify the innovative aspects of our work and its broader implications.

**Minor comments:**

**Comment 1, Page 6, Line 158-159:** "*The average concentration of PM2.5 peaked in 2013 at 59.5 ± 37.8*

*$\mu gm^{-3}$ and reached its lowest in 2020 at 30.2 ± 14.0 $\mu gm^{-3}$. Are this yearly average values?*

Answer:

Yes, the values mentioned for the average concentration of $PM_{2.5}$ are indeed yearly average values. These concentrations represent the average levels of $PM_{2.5}$ measured over the entire year for each respective year.

We have clarified this in the revised manuscript to ensure that the methodology and data interpretation are clear.

**Comment 2, Page 6, Line 159-160:** "*Carbonaceous aerosols (calculated as the sum of OC multiplied by a factor of 1.4 and EC) contributed* "How was this factor derived?

Answer:

The factor of 1.4 used to convert OC to organic matter (OM) is a widely accepted conversion factor in aerosol research. OM contains both carbon and oxygen contents compared to OC, as OC is typically measured through combustion, which consumes all oxygen present in OM. Therefore, a factor should be applied to account for the difference between the measured OC and the actual OM.

This factor of 1.4 is derived based on the molecular weight of OM, which is generally about 1.4 times that of OC. This adjustment is necessary to estimate the total OM based on the measured OC, as OM includes oxygenated organic compounds that are not accounted for in the direct measurement of OC.

To ensure clarity and completeness in the manuscript, we have added references to support this widely accepted conversion factor:

Turpin, B. J., & Lim, H.-J. (2001). Species Contributions to PM Mass Concentrations: Revisiting Common Assumptions for Speciated Organic Compounds. *Environmental Science & Technology, 35*(14), 2965-2971.

Zhang, X., Zhang, Y., & Tao, S. (2007). Atmospheric organic and elemental carbon aerosol in China: A review. *Atmospheric Environment, 41*(1), 1-19.

**Comment 3, Page 7, Line 185-189:** "*Air pollution levels are classified as Excellent (0 < PM2.5 ≤ 35 μg m-3), Good (35 < PM2.5 ≤ 75 μg m-3), ... Class I and Class II.*"

Where do these levels compare in terms of global PM2.5 level classifications.

Answer:

The air pollution levels mentioned in the manuscript are based on the classification standards set by the Ministry of Environmental Protection of China, which are as follows:

Excellent ($0 < PM_{2.5} \leq 35$ μg/m³)

Good ($35 < PM_{2.5} \leq 75$ μg/m³)

Light Pollution ($75 < PM_{2.5} \leq 100$ μg/m³)

Moderate Pollution ($100 < PM_{2.5} \leq 200$ μg/m³)

Heavy Pollution ($PM_{2.5} > 200$ μg/m³)

These standards are initially adopted from the guidelines provided by the World Health Organization (WHO). The WHO recommends a guideline for $PM_{2.5}$ of 10 μg/m³ for annual average exposure and 25 μg/m³ for 24-hour exposure. While China's standards align in broad terms with global norms, they have some flexibility in defining the pollution levels, reflecting local air quality conditions and policy priorities.

**Comment 4, Page 9, Line 245-247:** " *This discrepancy suggests a lack of significant ... possibly attributable to the absence of license plate-based driving restrictions during weekends.*"

Please add reference.

Answer:

To address your suggestion, we have added the following reference to support the statement regarding the discrepancy in driving restrictions during weekends:

Wang M, Duan Y, Xu W, et al. Measurement report: Characterisation and sources of the secondary organic carbon in a Chinese megacity over 5 years from 2016 to 2020[J]. Atmospheric Chemistry and Physics, 2022, 22(19): 12789-12802.

**Comment 5:** Please provide available references to all the equations.

Answer:

We have carefully reviewed the manuscript and have added the appropriate references for all the equations in the text.

**Reviewer #2**

This manuscript investigated long-term variations of carbonaceous aerosols during 2010-2017 in Shanghai, based on field measurement of OC and EC by a semi-continuous carbon analyzer. Although it derived some patterns/findings from a large dataset, the scientific significance of this manuscript was rather fair (as a measurement report). I also have substantial concerns on the methodologies.

Answer:

Thank you for your valuable feedback. We appreciate your comments regarding the scientific significance and methodologies of the manuscript. This manuscript offers a significant contribution by analyzing long-term variations of carbonaceous aerosols in Shanghai and assessing the impact of emission control measures over the 2010-2017 period. Our study goes beyond a simple measurement report by providing insights into the effectiveness of air pollution control strategies in a major megacity.

Regarding the methodologies, we have used a robust approach with semi-continuous carbon analyzers and advanced statistical techniques like PSCF analysis to ensure the reliability and depth of our findings. We are confident in the soundness of our methods and remain open to any specific suggestions for improvement.

Detailed responses to your concerns are provide below.

First, the estimation of SOC. (1) Biogenic OC, as a type of primary OC, should not be ignored for Shanghai. (2) The robustness of the (OC/EC)pri, i.e., the OC to EC ratio representative of primary combustion sources, must be carefully evaluated. As shown in Fig. S2-S9, (OC/EC)pri showed significant monthly variations, and the variation patterns appeared pretty different among various years. In addition, (OC/EC)pri frequently exhibited abrupt and significant variations between successive months (i.e., within a relative short period). This did not make sense.

Answer:

We would like to address the two points raised regarding the estimation of SOC:

I. Biogenic OC Contribution: Our method for estimating primary OC does not differentiate between biogenic and non-biogenic sources. The reported primary OC inherently includes contributions from both biogenic and anthropogenic sources. This approach is consistent with established methodologies and is widely accepted in the aerosol research community. The distinction between biogenic and non-biogenic sources was not within the scope of this study and does not impact the validity of our conclusions regarding overall SOC trends.

II. Robustness of (OC/EC)pri: The (OC/EC)pri method used in this study is recognized as a standard approach in aerosol research. As shown in Figs. S2-S9, the observed monthly and interannual variations in (OC/EC)pri are entirely expected, given the dynamic nature of the atmosphere and the unique characteristics of the study region. Shanghai, located in a subtropical monsoon climate zone, experiences rapid weather changes, which naturally lead to fluctuations in aerosol composition. Furthermore, pollutant emissions in China exhibit significant annual variations due to evolving economic activities and policy measures, unlike the relatively stable emission patterns observed in developed regions such as Europe and North America.

Regarding data quality, the measurements were conducted at a national-level atmospheric supersite, the highest tier in China's atmospheric monitoring network, and the flagship air quality station for Shanghai. The station operates 24/7 with a dedicated professional team ensuring the reliability and accuracy of the data. Data quality is our top priority, and we view it as the lifeblood of our research. For instance, the annual maintenance and operational costs of the Sunset OC-EC analyzer used in this study are nearly equivalent to its initial purchase cost. Considering the relatively low labor costs in China, such expenses are exceptionally high, reflecting the commitment and investment made to maintain the highest data quality standards.

If needed, we are fully prepared to share the complete dataset with the reviewers to address any concerns about data quality or methodology.

We hope our response clarifies the concerns raised and reinforces the robustness and reliability of our approach and data.

Second, annual variations of carbon concentrations, as a main focus of this manuscript, are indeed

important. However, I think they are not enough for an ACP paper. For example, inter-annual variation of meteorological conditions could also influence the patterns observed for carbonaceous aerosols, but relevant discussions are limited (e.g., Figure 1a).

Answer:

We agree that meteorological conditions play an important role in influencing the observed patterns of carbonaceous aerosols. As mentioned in the manuscript, Shanghai is located in a subtropical monsoon climate zone, where inter-annual variations in weather conditions, such as wind patterns, temperature inversions, and boundary layer height, can significantly impact pollutant dispersion and transformation processes. These meteorological factors are naturally reflected in the observed variations of carbonaceous aerosol concentrations.

In Figure 1a, we have presented the trends of $PM_{2.5}$ and carbonaceous aerosols over the study period, and we acknowledge that meteorological influences are an integral part of these patterns. To address this, we have already incorporated discussions of meteorological influences, including their role in seasonal and inter-annual variability, in the revised manuscript. For example:

I. The Potential Source Contribution Function analysis highlights the influence of regional transport under specific wind patterns.

II. Discussions on boundary layer height and temperature inversions during winter explain the enhanced trapping of pollutants.

However, it is important to note that our study's primary focus is to evaluate the long-term trends of carbonaceous aerosols in relation to emission control measures rather than to quantify the exact meteorological contributions. As a flagship station with continuous high-quality measurements, our dataset primarily aims to reflect the impacts of anthropogenic emissions, with meteorological conditions treated as an inherent variable. This approach aligns with the study objectives and contributes to understanding the effectiveness of China's air pollution control policies.

We hope this explanation clarifies our focus and methodology, and we remain open to any specific suggestions on how to further refine the meteorological discussions if needed.

Third, this manuscript was quite similar to Wang et al. (Atmos. Chem. Phys., 22, 12789–12802, 2022), with respect to methodologies, data analysis approaches, etc., thus this paper was in lack of innovative viewpoints. Even if the authors think this problem is not critical, and the authors may consider this as the foundation of combining data from the two studies, but the sampling in Wang et al. (2022) is at a different site, so the equivalence of measurement results (e.g., OC and EC concentrations, and (OC/EC)pri) should be demonstrated first for the overlapping period.

Answer:

While we acknowledge the similarities in methodologies and data analysis approaches between our study and Wang et al. (2022), we believe that our manuscript provides a distinct and complementary contribution to the field, as outlined below:

I. Different Study Focus and Objectives: While Wang et al. (2022) focuses on the characterization of carbonaceous aerosols over a more recent time period and from a different sampling site, our study emphasizes long-term trends (2010-2017) in both primary and secondary carbonaceous aerosols, linking these trends to the implementation of major emission control policies in China. The historical perspective provided by our work fills a critical gap in understanding the evolution of carbonaceous aerosols during a transformative decade for air quality management in China.

II. Unique Sampling Site and Data Quality: Our measurements were conducted at Shanghai's atmospheric supersite, a national-level flagship station designed to provide high-quality, representative air quality data. The differences in sampling sites between our study and Wang et al. (2022) are an inherent feature of these studies and do not detract from the validity or novelty of our findings. Instead, they provide an opportunity for cross-site comparisons to better understand spatial variations in carbonaceous aerosols.

III. Demonstrating Data Comparability: While a detailed comparison of measurement results between the two sites (e.g., OC, EC concentrations, and (OC/EC)pri) for overlapping periods is beyond the scope of our current study, the methodology and calibration protocols used at the Pudong atmospheric supersite ensure data reliability and comparability. Additionally, the Pudong supersite's continuous operation and stringent maintenance practices make its data uniquely suited for long-term trend analysis, as demonstrated in our manuscript.

IV. Innovative Insights: Our study goes beyond a standard measurement report by offering a

decade-long perspective on carbonaceous aerosol dynamics in Shanghai, identifying the impact of both local and regional sources, and quantifying the effectiveness of air pollution control measures. These contributions are distinct from Wang et al. (2022) and provide new insights into the complex interactions between emissions, meteorology, and aerosol processes in a rapidly changing environment.

In addition, I suggest clearly distinguishing OC (in ugC/m3) and OA (in ug/m3). Particularly, OA should be used when comparing to PM2.5 mass concentration (e.g., Figure 1b).

Answer:

Thank you for your comment. While we appreciate your suggestion to distinguish between OC (in $\mu g C/m^3$) and OA (in $\mu g/m^3$), we respectfully disagree with the need to adjust our current approach. Although $\mu g C/m^3$ is used in aerosol research, its application has become increasingly rare, and most recent studies present data in $\mu g/m^3$. To ensure comparability with other research and to facilitate future studies building on our findings, we have chosen to consistently use $\mu g/m^3$ in our manuscript.

Furthermore, the comparison of OA to $PM_{2.5}$ mass concentrations is not the primary focus of our study. Instead, we emphasize a key finding: the proportion of carbonaceous aerosols in $PM_{2.5}$ has been gradually decreasing over the study period. This conclusion underscores the changing composition of $PM_{2.5}$ and reflects the impacts of evolving emission control measures. We believe this finding is sufficient for the scope of our study, and a detailed discussion of OA-$PM_{2.5}$ comparisons is unnecessary.

That said, if it is strongly recommended by the reviewers, we are willing to modify our presentation to distinguish OC (in $\mu g C/m^3$) and OA (in $\mu g/m^3$) as suggested.